# Emergent Temporal Correspondences from Video Diffusion Transformers

**Jisu Nam**[*1]     **Soowon Son**[*1]     **Dahyun Chung**[2]     **Jiyoung Kim**[1]

**Siyoon Jin**[1]     **Junhwa Hur**[†3]     **Seungryong Kim**[†1]

[1]KAIST     [2]Korea University     [3]Google DeepMind

https://cvlab-kaist.github.io/DiffTrack

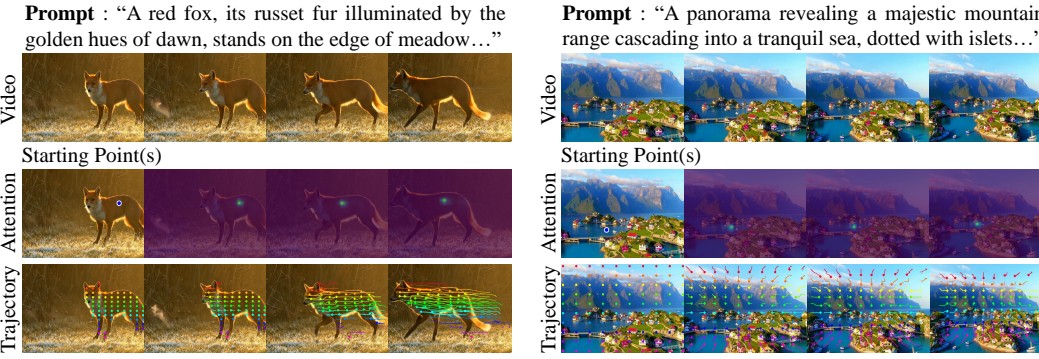

Figure 1: **Teaser:** DiffTrack reveals how video Diffusion Transformers (DiTs) establish temporal correspondences during video generation. Given a prompt and starting points, DiffTrack tracks how individual points align across subsequent frames via cross-frame attention in video DiTs (second row). This enables the extraction of coherent motion trajectories (third row) from both generated and real-world videos in a zero-shot manner.

## Abstract

Recent advancements in video diffusion models based on Diffusion Transformers (DiTs) have achieved remarkable success in generating temporally coherent videos. Yet, a fundamental question persists: how do these models internally establish and represent temporal correspondences across frames? We introduce *DiffTrack*, the first quantitative analysis framework designed to answer this question. DiffTrack constructs a dataset of prompt-generated video with pseudo ground-truth tracking annotations and proposes novel evaluation metrics to systematically analyze how each component within the full 3D attention mechanism of DiTs (*e.g.*, representations, layers, and timesteps) contributes to establishing temporal correspondences. Our analysis reveals that query-key similarities in specific, but not all, layers play a critical role in temporal matching, and that this matching becomes increasingly prominent during the denoising process. We demonstrate practical applications of DiffTrack in zero-shot point tracking, where it achieves state-of-the-art performance compared to existing vision foundation and self-supervised video models. Further, we extend our findings to motion-enhanced video generation with a novel guidance method that improves temporal consistency of generated videos without additional training. We believe our work offers crucial insights into the inner workings of video DiTs and establishes a foundation for further research and applications leveraging their temporal understanding.

---

[*]Equal contribution.

[†]Co-corresponding author.

39th Conference on Neural Information Processing Systems (NeurIPS 2025).

# 1 Introduction

Recent video diffusion models [25, 28, 49, 50, 54, 63, 69, 84, 88], powered by Diffusion Transformers (DiTs) [21, 61], have achieved remarkable progress in generating realistic videos with high photometric fidelity, temporal coherence, and complex motion. DiTs utilize full 3D attention to process all frames and text in a single sequence, enabling the effective propagation of spatial and temporal information, thus improving temporal coherence in generated videos. Their strong temporal priors have also led to various downstream tasks, such as video depth estimation [87], pose estimation [9], and 3D/4D reconstruction [23, 53, 74, 80].

Despite the success of these models, a fundamental question remains unanswered: *How do video DiTs establish temporal correspondences between frames during generation, and how can we explicitly extract them?* While existing works [31, 42, 56, 57, 75, 86] explore internal representations in U-Net-based image diffusion models, they mainly focus on two-frame correspondences.

In this paper, we introduce *DiffTrack*, the first in-depth quantitative analysis framework designed to pinpoint temporal matching within the video DiT architecture. DiffTrack provides insights into how and where video DiTs establish temporal correspondences during video generation, enabling the direct extraction of motion information from generated videos (*cf.* Fig. 1). Notably, our framework is adaptable to any DiT-based video generative models.

Our analysis investigates 3D attention, a core component of video DiTs, to determine which representations (*e.g.* query-key similarities *vs.* intermediate features), layers, and denoising timesteps are most critical for establishing temporal correspondence among frames.

To systematically analyze their intricate interplays during video generation, we construct a dataset of prompt-generated video using a video backbone under analysis and obtain pseudo ground-truth motion tracks from an off-the-shelf tracking method [44]. We then propose novel evaluation metrics that jointly assess temporal matching accuracy and confidence. Given the dataset, we extract descriptors at desired layers and timesteps, estimate temporal correspondence, and evaluate these correspondences using our proposed metrics.

Through the analysis, we uncover several key findings. 1) Query-key matching in 3D attention blocks provides clear temporal matching information. 2) A few specific layers play a dominant role in establishing temporal matching. 3) Temporal matching strengthens throughout the denoising process.

We further demonstrate the practical value of our analysis through experiments in zero-shot point tracking. By using the most significant feature descriptors for temporal matching, extracted at the optimal layer and timestep identified by DiffTrack, our approach achieves state-of-the-art results compared to existing vision foundation models [10, 18, 60, 68, 75] and self-supervised video models [3, 6, 40, 52, 65, 83], demonstrating the effectiveness of our framework.

Additionally, we extend our analysis to motion-enhanced video generation with a novel guidance technique, Cross-Attention Guidance (CAG). CAG works by perturbing cross-frame attention maps in the most dominant layers identified by DiffTrack, and guides the model away from degraded samples during sampling. Notably, CAG achieves significant gains in both human evaluations and automatic metrics, without requiring additional training, auxiliary networks, or supervision.

In summary, our contributions are:

- We identify the importance of understanding temporal correspondence in video DiTs and introduce DiffTrack, a novel framework that quantitatively analyzes and identifies temporal matching information within DiTs during video generation.

- We provide a detailed analysis of the open-source video DiT models including CogVideoX [84], HunyuanVideo [49], and CogVideoX-I2V [84], revealing key insights into their internal mechanisms.

- We demonstrate the effectiveness of DiffTrack in zero-shot point tracking, achieving state-of-the-art performance among existing vision foundation and self-supervised video models.

- We present motion-enhanced video generation with CAG, a novel guidance method that improves the motion consistency of generated videos without auxiliary models or supervision.

## 2 Preliminaries

### 2.1 Video Diffusion Models

Video diffusion models [25, 28, 49, 50, 54, 63, 69, 84, 88] generate videos from input text prompts through iterative denoising. To enhance efficiency, latent video diffusion models [30] perform this denoising in a latent space, typically using 3D Variational Autoencoders (VAE) [84] for spatial and temporal compression. A 3D VAE encodes a video $\mathbf{X} \in \mathbb{R}^{(1+F) \times H \times W \times 3}$, with its frame count $(1+F)$, height $(H)$, width $(W)$, into a compressed latent representation. This latent is then patchified and unfolded into a sequence $\mathbf{z}_{\text{video}}$ of length $(1+f)hw$, where $f$, $h$, and $w$ are derived from spatial compression ratio $p$ (*i.e.*, $h=H/p$ and $w=W/p$) and a temporal compression ratio $q$ (*i.e.*, $f=F/q$) with often skipping the first frame [84]. To incorporate text input, a text encoder [67] embeds the prompt into the text embedding $\mathbf{z}_{\text{text}}$ with sequence length $S$. Then the concatenated embeddings ($\mathbf{z}_{\text{video}}$ and $\mathbf{z}_{\text{text}}$), of length $(1 + f)hw + S$, guide the denoising process, enabling text-aligned video generation. Given a predetermined noise schedule, forward process progressively adds Gaussian noise to $\mathbf{z}_{\text{video},t}$ for timestep $t$, producing noisier latents $\mathbf{z}_{\text{video},t+1}$. In the denoising process, a neural network $\epsilon_\theta(\mathbf{z}_{\text{video},t}, \mathbf{z}_{\text{text}}, t)$ predicts and removes the noise to obtain $\mathbf{z}_{\text{video},t-1}$, aligning with the text prompt. After $T$ denoising steps, the fully denoised latent $\mathbf{z}_{\text{video},0}$ is decoded by the 3D VAE to generate the final video $\mathbf{X}'$.

### 2.2 Video Diffusion Transformers

Following the success of Sora [54], DiT [21, 61] has become a standard architecture for video generative models [25, 28, 49, 50, 84]. DiT employs multiple layers of full 3D attention, enabling direct interaction between visual and textual information through attention and feed-forward processing.

At each timestep $t$, the concatenated sequences of $\mathbf{z}_{\text{video},t}$ and $\mathbf{z}_{\text{text}}$ are augmented with 3D positional embeddings, *e.g.*, RoPE [73] or sinusoidal embeddings [79]. Then, at each layer $l$, full 3D attention transforms the sequences into queries $\mathbf{Q}_{t,l}$, keys $\mathbf{K}_{t,l}$, and values $\mathbf{V}_{t,l}$, all lying in $\mathbb{R}^{((1+f)hw+S) \times d}$ with the channel dimension $d$. The resulting output of full 3D attention is computed as:

$$\text{Attn}(\mathbf{Q}_{t,l}, \mathbf{K}_{t,l}, \mathbf{V}_{t,l}) = \mathbf{A}_{t,l} \mathbf{V}_{t,l} \tag{1}$$

$$\text{with} \quad \mathbf{A}_{t,l} = \text{Softmax}(\mathbf{Q}_{t,l} \mathbf{K}_{t,l}^T / \sqrt{d}). \tag{2}$$

To understand how the video latent $\mathbf{z}_{\text{video},t}$ and text embedding $\mathbf{z}_{\text{text}}$ interact in full 3D attention, the notations of query $\mathbf{Q}_{t,l}$ and key $\mathbf{K}_{t,l}$ can be rewritten at the token sequence level:

$$\mathbf{Q}_{t,l} = \text{Concat}(\mathbf{Q}_{t,l}^1, \ldots, \mathbf{Q}_{t,l}^{1+f}, \mathbf{Q}_{t,l}^{\text{text}}) \tag{3}$$

$$\mathbf{K}_{t,l} = \text{Concat}(\mathbf{K}_{t,l}^1, \ldots, \mathbf{K}_{t,l}^{1+f}, \mathbf{K}_{t,l}^{\text{text}}). \tag{4}$$

Here, $\mathbf{Q}_{t,l}^i$ and $\mathbf{K}_{t,l}^i$ are projections from the $i$-th frame latent, with $i \in \{1, \ldots, 1+f\}$, and $\mathbf{Q}_{t,l}^{\text{text}}$ and $\mathbf{K}_{t,l}^{\text{text}}$ are from the text embeddings. $\text{Concat}(\cdot)$ concatenates these across the token sequences.

Fig. 2 illustrates full 3D attention in video DiTs. It can be categorized into four key interactions: 1) self-frame attention $\mathbf{A}_{t,l}^i$, 2) cross-frame attention $\mathbf{A}_{t,l}^{i,j}$, 3) text-frame attention $\mathbf{A}_{t,l}^{\text{text},j}$ or $\mathbf{A}_{t,l}^{i,\text{text}}$, and 4) self-text attention $\mathbf{A}_{t,l}^{\text{text}}$, where $i, j \in [1, 1+f]$ with $i \neq j$. Self-frame attention attends to its own frame, focusing on nearby spatial patches at the pixel location. Text-frame attention injects semantic information from text embeddings into the frame latents. Self-text attention refines the textual context itself.

Of these, cross-frame attention $\mathbf{A}_{t,l}^{i,j} \in \mathbb{R}^{hw \times hw}$ is of particular interest to our study. Its ability to allow every pixel in one frame to interact with any pixel in another frame is what explicitly enables the model to build and understand temporal relationships across the video sequence.

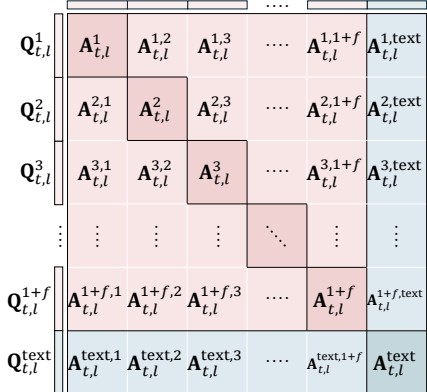

Figure 2: **Illustration of full 3D attention in video DiTs**, where video frame latents and text embeddings interact.

Starting Points                                    Starting Points

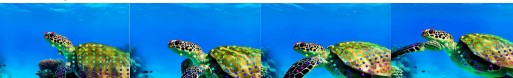

**Prompt** : "A curious sea turtle, its shell dappled with hues of emerald and bronze, drifts gracefully through the clear blue waters of a vibrant coral reef. Sunlight filters through the gentle waves, …"

**Prompt** : "A breathtaking aerial view captures the light of dawn as it spills over mountain peaks, casting long shadows across the rugged terrain. The drone glides smoothly, revealing a tapestry of colors…"

(a) Object Dataset                                    (b) Scene Dataset

Figure 3: **Our curated evaluation dataset** includes: (a) an object dataset for dynamic object-centric videos and (b) a scene dataset for static scenes with camera motion. Each dataset comprises 50 prompt-generated video pairs per video generative model (*e.g.* CogVideoX-2B [84]). In the benchmark, we predefine starting points in the first frame and obtain pseudo ground-truth trajectories using an off-the-shelf tracking method [44].

## 3   DiffTrack

We introduce *DiffTrack*, a framework to quantify how video DiTs capture temporal correspondence during video generation. This is crucial yet nontrivial due to the inherent complexity of video DiTs, which involve multiple layers, denoising timesteps, and full 3D attention. To systematically analyze them, DiffTrack provides an evaluation dataset and novel metrics for assessing the accuracy, confidence, and influence of temporal matching. While we present an in-depth analysis of the open-source video DiT model, CogVideoX-2B [84], our framework is broadly applicable to other DiT architectures, as demonstrated by our analyses of CogVideoX-5B [84], HunyuanVideo [49] and CogVideoX-5B-I2V [84] in Sec. B.

### 3.1   Evaluation Dataset

To accurately investigate temporal correspondence during video generation, it's crucial to simulate the generation process faithfully. Reconstructing real-world videos often introduces challenges such as inversion errors [71] due to imprecise prompts and distribution shifts from training data. To circumvent this, we curate specialized evaluation prompts and generate corresponding video datasets using the specific backbone under analysis. This approach allows perfect video reconstruction during evaluation, as we use paired prompts with fixed seeds. For an analysis using real-world videos from DAVIS [64], please refer to Sec. E.

To analyze videos with object and camera motion, we curate two distinct datasets: (1) an object dataset for dynamic object-centric videos and (2) a scene dataset for static scenes with camera motion. Each dataset includes 50 text prompts with corresponding 50 videos (*e.g.* $480 \times 720$ resolution, 49 frames, generated by CogVideoX-2B [84]).

To assess temporal matching, the datasets also include pseudo ground truth track annotations. We predefine a set of starting points $\mathbf{p}^1 \in \mathbb{R}^{N \times 2}$ (in latent resolution) in the first frame of each video, where $N$ is the number of points. For the object dataset, we use SAM [48] to segment foreground objects and sample grid points with a spacing of $1/20$ of the video's spatial resolution. For the scene dataset, we sample a $10 \times 10$ regular grid of points across the entire frame. Fig. 3 shows examples of each dataset; further details are provided in Sec. A. Since ground-truth tracks are unavailable for generated videos, we use an off-the-shelf tracking method, CoTracker [44], to obtain pseudo ground truth $\mathbf{T} \in \mathbb{R}^{F \times N \times 2}$ (in original resolution).

### 3.2   Temporal Correspondence Estimation within Video DiTs

To evaluate matching accuracy, we first extract temporal correspondences across video frames using feature descriptors from a video model. We independently establish pairwise correspondences between the first frame and each subsequent frame.

We first construct a matching cost between the descriptors from the first latent and each $j$-th latent, with $j \in \{2, \ldots, 1+f\}$, at each timestep $t$ and layer $l$:

$$\mathbf{C}_{t,l}^{1,j} = \texttt{Softmax}(\mathbf{D}_{t,l}^1 (\mathbf{D}_{t,l}^j)^T / \sqrt{d}), \tag{5}$$

where $\mathbf{D}_{t,l}^1$ and $\mathbf{D}_{t,l}^j$ are the feature descriptors corresponding to the first frame latent and the $j$-th latent, and $d$ is the channel dimension of both $\mathbf{D}_{t,l}^1$ and $\mathbf{D}_{t,l}^j$. $\texttt{Softmax}(\cdot)$ is applied over the keys for each pixel in the query. Descriptors $\mathbf{D}_{t,l}$ are internal representation candidates from video DiTs, such as intermediate features after attention blocks or query-key matrices within attention blocks.

Matched correspondence points $\mathbf{p}_{t,l}^j \in \mathbb{R}^{N \times 2}$ at the $j$-th frame latent, timestep $t$, and layer $l$, with the number of points $N$, are obtained through $\texttt{Argmax}$, which identifies the spatial location $\mathbf{x}$ of the highest value in $\mathbf{C}_{t,l}^{1,j}$ within the spatial domain of the $j$-th latent $\Omega$.

$$\mathbf{p}_{t,l}^j = \underset{\mathbf{x} \in \Omega}{\texttt{Argmax}} \ \mathbf{C}_{t,l}^{1,j}(\mathbf{p}^1, \mathbf{x}). \tag{6}$$

Motion tracks along video frames are obtained by concatenating ($\texttt{Concat}$) the starting point $\mathbf{p}^1$ with the estimated matches $\mathbf{p}_{t,l}^j$ across the video latent space. These tracks, in the latent spatial coordinates, are then spatio-temporally upscaled to the original RGB coordinates through linear interpolation ($\texttt{Interp}$), yielding $\hat{\mathbf{T}}_{t,l} \in \mathbb{R}^{F \times N \times 2}$:

$$\hat{\mathbf{T}}_{t,l} = \texttt{Interp}(\texttt{Concat}(\mathbf{p}^1, \mathbf{p}_{t,l}^2, \ldots, \mathbf{p}_{t,l}^{1+f})). \tag{7}$$

### 3.3 Evaluation Metrics

Given the estimated matched points, we propose three complementary metrics for evaluating temporal matching in video generation: *matching accuracy*, *confidence score*, and *attention score*. Specifically, matching accuracy measures the precision of estimated tracks across frames. The confidence score quantifies the certainty with which the starting point attends to its estimated match. The attention score reflects the relative strength of cross-frame attention during generation, compared to self-frame and text-frame attention.

**Matching Accuracy.** We evaluate point accuracy using the percentage of correct keypoints (PCK) with a predefined error threshold between the estimated track $\hat{\mathbf{T}}_{t,l}$ and visible points in the ground truth $\mathbf{T}$. The *matching accuracy* averages PCK over all visible points across cross-latents and videos.

**Confidence Score.** We use the maximum attention values between the first and $j$-th frame latent $\mathbf{A}_{t,l}^{1,j}$ (*cf.* Sec. 2.2) to measure how confidently the starting points $\mathbf{p}^1$ attend to their predicted matched points, formulated as:

$$\mathbf{M}_{t,l}^j = \underset{\mathbf{x} \in \Omega}{\texttt{Max}} \ \mathbf{A}_{t,l}^{1,j}(\mathbf{p}^1, \mathbf{x}), \tag{8}$$

which quantifies the attention of $\mathbf{p}^1$ to the estimated point in the $j$-th latent at timestep $t$ and layer $l$. $\texttt{Max}$ takes the maximum attention value over all spatial locations $\mathbf{x} \in \Omega$. The *confidence score* is the average of these maximum values across all cross-latents with $j \in \{2, \ldots, 1+f\}$, all visible points in ground truth, and videos.

**Attention Score.** We use the sum of cross-attention values across all cross-frame latents. This allows us to assess the influence of cross-frame interactions during video generation, in comparison to other types of attention, such as text-frame or self-frame attention. This is formulated as:

$$\mathbf{S}_{t,l} = \sum_{j \in \mathcal{F}} \sum_{\mathbf{x} \in \Omega} \mathbf{A}_{t,l}^{1,j}(\mathbf{p}^1, \mathbf{x}), \tag{9}$$

which quantifies the sum of attention from $\mathbf{p}^1$ across cross-frames at timestep $t$ and layer $l$, over all spatial locations $\mathbf{x} \in \Omega$ and all cross-frame indices $j \in \mathcal{F}$, where $\mathcal{F} = [2, 1+f]$. The *attention score* is the average of these summation values across all visible points in the ground truth and all videos.

Three metrics must be considered together (as detailed in Sec. 3.4), as none alone is sufficient to ensure temporal matching. For instance, even with high matching accuracy across frames, it may not influence the generation process when attention scores from others (*e.g.*, self-frame or text-frame) are higher. In another case, high confidence indicates the certainty of the matching score but does not ensure the correctness of the match. We thus compute the harmonic mean of the normalized matching accuracy, confidence score, and attention score to identify instances where all three metrics are high.

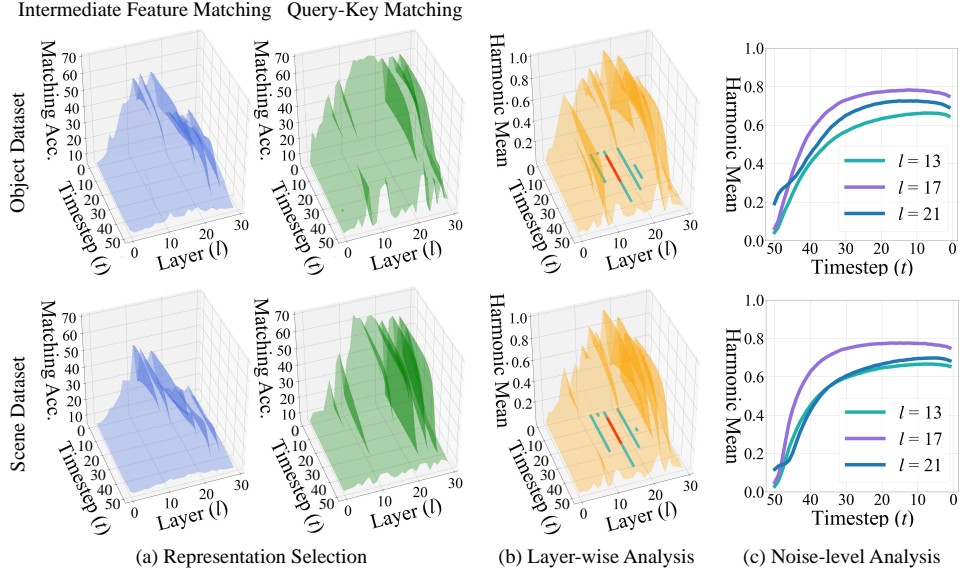

Figure 4: **Analysis of temporal matching in CogVideoX-2B [84].** (a) Query-key matching outperforms intermediate feature matching, highlighting the effectiveness of cross-frame interactions in 3D attention. (b) The harmonic mean of query-key matching shows that temporal matching is primarily driven by a few specific layers. (c) Temporal matching improves progressively during the denoising but slightly degrades near the final steps.

## 3.4 Analysis

With DiffTrack, we systematically analyze CogVideoX-2B [84] in the context of temporal correspondence. For all analyses, we use full 3D attention in the model, which consists of 30 layers with 50 denoising timesteps.

Our analysis considers the following three perspectives. *Representation selection* compares intermediate features and query-key representations. *Layer-wise analysis* explores how well temporal matching is encoded at different depths within the attention blocks. *Noise-level analysis* examines how temporal relationships evolve throughout the denoising process. Further in-depth analysis is provided in Sec. E. Additional analyses of CogVideoX-5B [84], HunyuanVideo [49] and CogVideoX-5B-I2V [84] are provided in Sec. B.

**Representation Selection.** Fig. 4(a) compares the accuracy of intermediate feature matching, where features are extracted after each attention layer, and query-key matching, where queries and keys are obtained within each attention layer. Our results indicate that *query-key matching consistently outperforms intermediate feature matching*. This finding aligns with prior works [3, 57], in which query-key matching captures geometric relationships for correspondence, while values contain visual appearance, potentially diluting geometric cues for accurate matching.

**Layer-wise Analysis.** Fig. 4(b) presents the harmonic mean of query-key matching across all timesteps and layers to identify which feature descriptors at which layer and timestep play a predominant role in temporal matching. We observe that the top-20 scores (red) originate from the same layer, indicating that *a specific layer predominantly governs temporal correspondence*. This behavior is further observed in the top-50 scores (green): a limited set of layers drives temporal matching.

**Noise-level Analysis.** Fig. 4(c) presents the harmonic mean of query-key matching across timesteps in the selected layers from Fig. 4(b), which are identified as leading layers for temporal matching. *Temporal matching improves during the denoising process but slightly degrades toward the end.* This is because earlier timesteps (1) contain noisier latents, which hinder precise temporal matching, and (2) rely heavily on text embeddings and self-frame attention to establish the overall video semantics and layout. Toward the final timesteps, (3) self- and text-frame at-

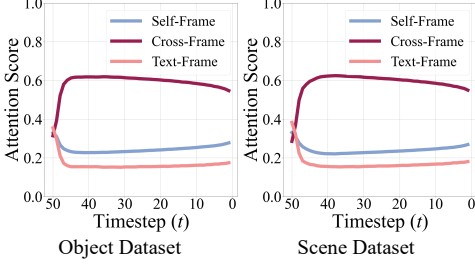

Figure 5: **Evolution of attention scores across timesteps.**

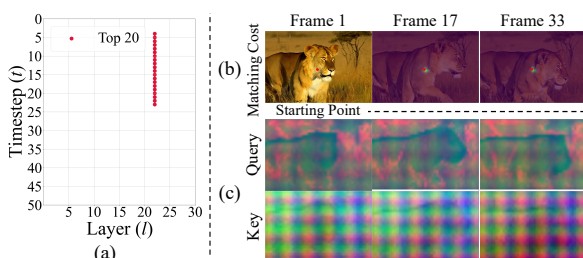

Figure 6: **Analysis of positional bias in temporal matching.**

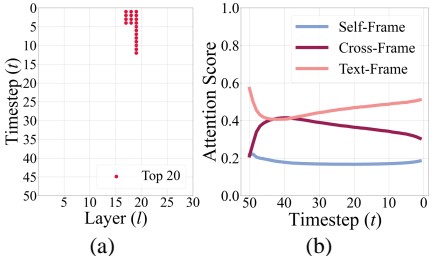

Figure 7: **Impact of persistent text-frame attention on attention score.**

tention slightly increase again to refine the remaining
appearance details, once the motion in the synthesized video has been established.

This observation is supported by Fig. 5, which shows the attention scores for text-frame, self-frame, and cross-frame attention along timesteps. After the early denoising steps, text-frame and self-frame attention remain low, reducing the influence of textual guidance. Meanwhile, cross-frame attention remains the most influential, enhancing cross-frame coherence, with a slight drop near the end of the timesteps.

**Additional Analysis on Metrics.** We further emphasize the importance of jointly considering three metrics, matching accuracy, confidence score, and attention score. Fig. 6(a) presents the top-20 timesteps and layers where the confidence score is high, but matching accuracy is low. We find that specific layers exhibit this discrepancy as they are overwhelmed by positional information induced by positional embeddings at each timestep. In Fig. 6(c), PCA visualization of queries and keys in these layers reveals a dominance of positional cues, while Fig. 6(b) shows that in these layers, matching cost visualizations indicate that points in the first frame tend to match exactly with their initial locations in other frames. This suggests that these points are not correctly matched to their actual counterparts but instead strongly attend to the same spatial location across frames, reflecting the impact of positional bias.

Additionally, Fig. 7(a) presents the top-20 timesteps and layers where matching accuracy and confidence scores are high, but attention scores are low. We observe that this property is exhibited in certain layers. As shown in Fig. 5, this occurs because text-frame attention remains highly active in these layers, maintaining a value around 0.5, unlike in other layers where text-frame attention drops below 0.2 (*cf.* Fig. 5). This reduces the attention scores, which in turn limits the influence of cross-frame interactions during the generation process.

## 4   DiffTrack for *Zero-Shot* Point Tracking

DiffTrack enables the joint extraction of motion trajectories and video generation, selecting the optimal layer and timestep based on matching accuracy. We demonstrate this in zero-shot point tracking [5] on real videos, without training specialized architectures [19] or fine-tuning video diffusion models [41]. To achieve this, we use the inverted noise of real videos obtained through DDIM inversion [71] at the selected timestep and extract features from the chosen layer. Notably, the inversion error is negligible, as we use the final timestep $t = 1$ based on our analysis of matching accuracy in Fig. 4(a). However, this still faces challenges such as temporal context loss from 3D VAE compression and handling of long-term video sequences. We address these challenges below. The overall architecture and its details are provided in Sec. C.

**Temporal Compression in 3D VAE.** As discussed in Sec. 2.1, the 3D VAE temporally compresses video frames into a single-frame latent with a compression ratio $q$. While linear interpolation can recover motion trajectories from the latent to the RGB video space, it often fails to capture per-frame motion details, limiting accuracy. To mitigate this, we set $q=1$ to establish a direct one-to-one mapping between each video frame and its latent, thereby avoiding temporal compression and enabling precise tracking. In Sec. E, we demonstrate that the one-to-one mapped latents can still reconstruct the original videos using the 3D VAE decoder.

**Long-term Video Sequences.** The fixed temporal resolution of pre-trained video models (*e.g.* 49 frames in CogVideoX-2B [84]) limits their ability to model long-term contexts. Naively splitting

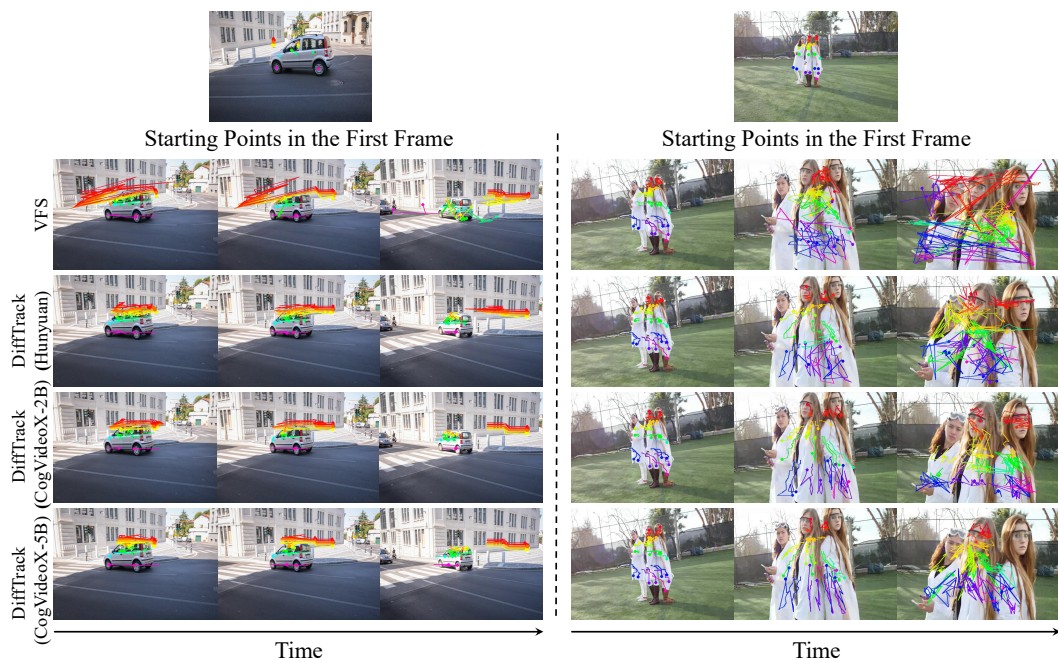

Figure 8: **Qualitative comparison.** CogVideoX-2B [84] combined with DiffTrack produces smoother and more accurate trajectories than DINOv2 [60] and VFS [83], which struggle with temporal dynamics and often yield inconsistent tracks.

| Backbone | Kinetics | | | | | | DAVIS | | | | | |
|---|---|---|---|---|---|---|---|---|---|---|---|---|
| | $< \delta^0$ | $< \delta^1$ | $< \delta^2$ | $< \delta^3$ | $< \delta^4$ | $< \delta^x_{avg}$ | $< \delta^0$ | $< \delta^1$ | $< \delta^2$ | $< \delta^3$ | $< \delta^4$ | $< \delta^x_{avg}$ |
| DINO (ViT-B/16) [10] | 2.8 | 10.9 | 33.9 | 58.2 | 74.8 | 36.1 | 2.9 | 10.7 | 34.3 | 59.8 | 77.0 | 37.3 |
| DINOv2-Reg (ViT-B/14) [18] | 2.8 | 10.8 | 32.8 | 60.6 | 78.7 | 37.1 | 3.0 | 11.7 | 36.5 | 67.5 | 84.1 | 40.6 |
| DINOv2 (ViT-B/14) [60] | 3.0 | 11.4 | 34.6 | 63.0 | 80.3 | 38.4 | 3.1 | 12.1 | 38.7 | 70.7 | **85.8** | 42.1 |
| DIFT (SD1.5) [75] | 3.7 | 14.6 | 44.6 | 69.0 | 77.5 | 41.9 | 3.5 | 13.0 | 39.3 | 63.1 | 72.2 | 38.2 |
| DIFT (SD2.1) [75] | 3.7 | 14.9 | 45.4 | 70.9 | 79.6 | 42.9 | 3.6 | 13.3 | 40.1 | 65.8 | 75.7 | 39.7 |
| SMTC (ViT-S/16) [65] | 4.1 | 15.5 | 34.2 | 54.4 | 72.1 | 36.1 | 2.6 | 12.1 | 29.4 | 52.5 | 73.0 | 33.9 |
| CRW (ResNet-18) [40] | 5.2 | 19.4 | 42.7 | 62.9 | 74.3 | 40.9 | 3.1 | 13.9 | 34.7 | 57.1 | 70.5 | 35.9 |
| Spa-then-Temp (ResNet-50) [52] | 5.3 | 19.4 | 41.6 | 58.9 | 69.7 | 39.0 | 3.2 | 13.8 | 33.1 | 53.4 | 67.5 | 34.2 |
| VFS (ResNet-50) [83] | 5.4 | 20.1 | 44.6 | 65.4 | 76.6 | 42.4 | 3.5 | 15.2 | 37.2 | 60.8 | 75.2 | 38.4 |
| SVD [6] | 4.3 | 16.0 | 37.9 | 56.3 | 69.8 | 36.6 | 3.6 | 14.6 | 34.1 | 55.7 | 71.4 | 35.9 |
| ZeroCo (CroCo) [3] | **14.5** | 22.9 | 35.9 | 60.4 | 79.7 | 42.6 | 4.6 | 8.8 | 19.5 | 44.9 | 65.6 | 28.7 |
| **DiffTrack (HunyuanVideo [49])** | 5.9 | 22.0 | 49.1 | 70.4 | 80.3 | 45.5 | 4.4 | 18.2 | 44.8 | 70.1 | 82.8 | 44.1 |
| **DiffTrack (CogVideoX-2B [84])** | 6.2 | 23.3 | 51.2 | 71.2 | 79.9 | 46.3 | 4.8 | 19.4 | 49.2 | 73.6 | 84.3 | 46.3 |
| **DiffTrack (CogVideoX-5B [84])** | 6.8 | **25.9** | **55.4** | **74.9** | **82.7** | **49.2** | 5.2 | **20.5** | **50.7** | **73.9** | 84.3 | **46.9** |

Table 1: **Quantitative comparison on the TAP-Vid datasets [19].** Video DiTs [49, 84] combined with DiffTrack outperform all vision foundation models trained on single images and self-supervised models trained on two-view images or videos for zero-shot tracking.

and processing video chunks separately disrupts direct temporal correspondence with the global first frame. To address this, we construct each chunk to include the global first frame, maintaining a direct temporal connection while interleaving subsequent frames to minimize large motion changes.

## 4.1 Experimental Settings

Implementation details and ablation studies are provided in Sec. C.2 and Sec. F.1. We evaluate zero-shot tracking on two real-video datasets with precisely annotated tracks: TAP-Vid-DAVIS [19] and TAP-Vid-Kinetics [19], following [46]. We measure the position accuracy of estimated tracks as the percentage of predicted points within thresholds from visible ground-truth points. We adopt five threshold levels [16, 19, 44, 46]: $\delta^0, \delta^1, \delta^2, \delta^3, \delta^4$, corresponding to pixel distances of 1, 2, 4, 8, and 16, respectively, and report the average accuracy across all thresholds as $\delta^x_{avg}$. Starting points are sampled from the first frame as in [5].

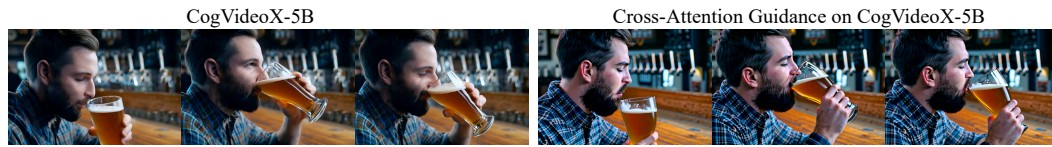

CogVideoX-5B  ·  Cross-Attention Guidance on CogVideoX-5B

**Prompt**: "… He takes a slow, appreciative sip, his eyes closing momentarily as he savors the complex flavors. …"

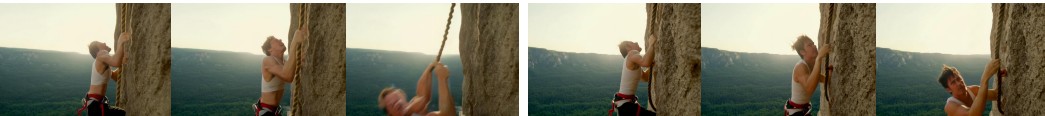

**Prompt**: "A determined individual, … ascends a thick, rugged rope hanging from a towering rock face. …"

Figure 10: **Qualitative comparison with baseline [84] and CAG.** CAG enhances temporal matching and corrects motion inconsistencies in the synthesized videos.

## 4.2 Experimental Results

We compare our method with existing vision foundation models trained on single images [10, 18, 60, 68, 75] and self-supervised models trained on two-view images [3] or videos [6, 40, 52, 65, 83]. As shown in Tab. 1, our approach achieves superior performance on both the Kinetics and DAVIS datasets, ultimately obtaining the highest average accuracy in $\delta^x_{\text{avg}}$. The results highlight our in-depth analysis of temporal matching within the full 3D attention mechanism of video DiTs.

Fig. 8 shows predicted motion trajectories on the DAVIS dataset, alongside qualitative comparisons between our method and prior approaches [60, 83]. Unlike previous methods, which struggle with temporal dynamics and often yield inconsistent tracks, DiffTrack on CogVideoX-2B [84] produces smoother and more accurate trajectories. Additional qualitative and quantitative results are provided in Sec. G and Sec. H.

## 5 DiffTrack for *Motion-Enhanced* Video Generation

We extend our findings to generate motion-enhanced videos by improving temporal correspondence within full 3D attention. As illustrated in Fig. 9, we propose *Cross-Attention Guidance (CAG)*, a novel diffusion guidance technique applied at specific layers identified in Fig. 4(b), steering video generation toward motion-enhanced samples. Unlike prior works [11, 41] that require large-scale video-trajectory training pairs, CAG requires no additional training, external conditions, or auxiliary modules, and operates entirely within the existing diffusion framework.

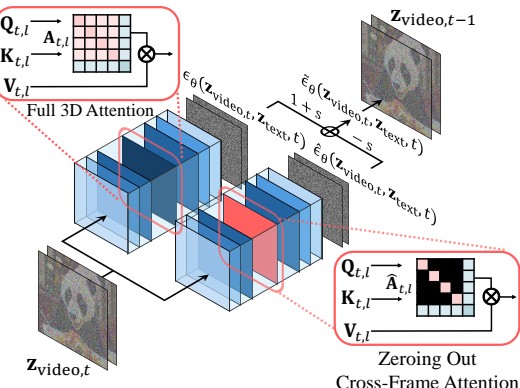

Figure 9: **Overall architecture of CAG.**

Inspired by PAG [2], CAG simulates degraded motion by zeroing out selected cross-frame attention maps (*e.g.* $l = 13, 17, 21$ in CogVideoX-2B, based on the harmonic mean in Fig. 4(b)), then guides the diffusion model away from these degraded samples, promoting temporally coherent video generation. This can be formulated as:

$$\tilde{\boldsymbol{\epsilon}}_\theta(\mathbf{z}_{\text{video},t}, \mathbf{z}_{\text{text}}, t) = \boldsymbol{\epsilon}_\theta(\mathbf{z}_{\text{video},t}, \mathbf{z}_{\text{text}}, t) + s \cdot (\boldsymbol{\epsilon}_\theta(\mathbf{z}_{\text{video},t}, \mathbf{z}_{\text{text}}, t) - \hat{\boldsymbol{\epsilon}}_\theta(\mathbf{z}_{\text{video},t}, \mathbf{z}_{\text{text}}, t)), \quad (10)$$

where $\boldsymbol{\epsilon}_\theta(\cdot)$ is the standard noise prediction at timestep $t$, conditioned on the text. $\hat{\boldsymbol{\epsilon}}_\theta(\cdot)$ denotes the noise prediction from a perturbed forward pass, where cross-frame attention maps $\mathbf{A}_{t,l}^{i,j}$ in selected layers are zeroed out to simulate motion degradation, producing $\hat{\mathbf{A}}_{t,l}$. $s$ is the guidance scale, and the final guided prediction $\tilde{\boldsymbol{\epsilon}}_\theta(\cdot)$ steers the model to denoise away from motion-degraded samples.

## 5.1 Experimental Settings

Further implementation and evaluation details are provided in Sec. D, and ablation studies are included in Sec. F.2. We evaluate CAG against its baselines, CogVideoX-2B and CogVideoX-5B, using both automatic metrics and human evaluation.

For automatic metrics, we report temporal quality metrics: Subject Consistency, Background Consistency, and Dynamic Degree from Vbench [38]. Subject Consistency and Background Consistency measure the temporal coherence of subject and background appearance, respectively. While static scenes can achieve high scores on these metrics, we additionally calculate Dynamic Degree to quantify motion dynamics. We also report a frame-wise quality metric from Vbench, Imaging Quality, which detects frame-wise distortions.

For human evaluation, we follow the Two-Alternative Forced Choice (2AFC) protocol [6, 11, 68], where each rater compares a video from the baseline with a video from our method (baseline + CAG), and selects one based on overall video quality, motion, and text-video alignment. We collect 750 responses from a total of 25 participants for each baseline.

## 5.2 Experimental Results

In Tab. 2, CAG outperforms the baseline across all human evaluations. CAG achieves higher scores in Subject Consistency, Background Consistency, Dynamic Degree, and Imaging Quality on VBench, indicating that our guidance improves motion dynamics while enhancing motion consistency and overall video fidelity.

| Method | Human Eval | | | Auto. Metrics | | | |
|---|---|---|---|---|---|---|---|
| | Video Quality | Motion Fidelity | Text Faithfulness | Subject Consistency | Background Consistency | Dynamic Degree | Imaging Quality |
| CogVideoX-2B | 12.85 | 14.99 | 18.56 | 0.9276 | 0.9490 | 0.7917 | 0.5657 |
| CogVideoX-2B + **CAG** | **87.16** | **85.01** | **81.44** | **0.9313** | **0.9564** | **0.8235** | **0.6054** |
| CogVideoX-5B | 39.10 | 40.91 | 30.00 | 0.9158 | 0.9590 | 0.6667 | 0.5531 |
| CogVideoX-5B + **CAG** | **60.90** | **59.09** | **70.00** | **0.9283** | **0.9644** | **0.6863** | **0.6051** |

Table 2: **Quantitative results of CAG.** Human evaluation reports the percentage of votes.

In Fig. 10, the baseline often fails to generate consistent and natural motion, resulting in physically implausible motion (*e.g.* drinking beer in the first row) or blurry and disjointed appearances (*e.g.* a blurred and fragmented human body in the second row). In contrast, CAG effectively corrects these artifacts by enhancing temporal matching and motion consistency, ensuring that corresponding physical points remain coherent across frames. More qualitative results are provided in Sec. G.

We further compare CAG with Spatiotemporal Skip Guidance (STG) [39], which also aims to enhance motion without additional training. STG degrades the original model by selectively skipping spatiotemporal layers (including self-frame and text-frame attention) and then uses the degraded model as guidance.

| Method | Auto. Metrics | | | |
|---|---|---|---|---|
| | Subject Consistency | Background Consistency | Dynamic Degree | Imaging Quality |
| CogVideoX-2B | 0.9276 | 0.9490 | 0.7917 | 0.5657 |
| CogVideoX-2B + **STG** | 0.9263 | 0.9507 | 0.7777 | 0.6031 |
| CogVideoX-2B + **CAG** | **0.9313** | **0.9564** | **0.8235** | **0.6054** |

Table 3: **Quantitative comparison between STG and CAG.**

As shown in Tab. 3, CAG outperforms STG across motion consistency, motion dynamics, and video quality. While STG modifies both self-frame and text-frame attention and often distorts scene layout or content, CAG only zeroes out cross-frame attention in dominant layers, enhancing motion while preserving the original spatial structure and semantics. Moreover, STG does not analyze how temporal correspondence emerges within video DiTs, thus requiring heuristic layer selection for each model, whereas CAG leverages DiffTrack-based analysis to identify optimal layers for motion-enhanced guidance.

## 6 Conclusion

We introduce DiffTrack, a framework for analyzing temporal correspondence in video DiTs, revealing how these models establish temporal correspondences during video generation. Our analysis identifies the crucial role of query-key similarities within specific layers of the full 3D attention mechanism and shows that temporal matching strengthens during denoising. We demonstrate the practicality of DiffTrack in zero-shot point tracking and motion-enhanced video generation, paving the way for leveraging temporal understanding in downstream tasks and improving video generation quality.

## Acknowledgments and Disclosure of Funding

This research was supported by Institute of Information & communications Technology Planning & Evaluation (IITP) grant funded by the Korea government (MSIT) (RS-2019-II190075, RS-2024-00509279, RS-2025-II212068, RS-2023-00227592, RS-2025- 02214479, RS-2024-00457882, RS-2025-25441838, RS-2025-25441838, RS-2025-02214479, RS-2025-02217259) and the Culture, Sports, and Tourism R&D Program through the Korea Creative Content Agency grant funded by the Ministry of Culture, Sports and Tourism (RS-2024-00345025, RS-2024-00333068, RS-2023-00222280, RS-2023-00266509), and National Research Foundation of Korea (RS-2024-00346597).

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

# Appendix

## A   Dataset Curation Details

In this section, we provide further details on the curation of our evaluation dataset. In Fig. A.13 and Fig. A.14, we present 50 prompts per dataset for analysis on CogVideoX-2B. Additionally, we provide more dataset examples in Fig. A.15.

**Curating Prompt-Video Pairs.**   To analyze motion trajectories in depth, we require high-quality prompt-video pairs to ensure motion consistency and video fidelity, as suboptimal videos could introduce noise into our analysis. To achieve this, we begin with collecting GPT-enhanced prompts from VBench [38], a benchmark designed for evaluating video generative models. For the object dataset, we gather prompts from animal categories, while for the scene dataset, we collect prompts from architecture and lifestyle categories. We further augment both datasets by generating additional prompts using GPT-4o [1], with VBench prompts as references, ultimately collecting 300 prompts for both the object and scene datasets. We then synthesize videos from each prompt using video generative models under analysis (*e.g.* CogVideoX-2B [84]). To simulate the generation process, we curate a distinct evaluation dataset for each analyzed model (CogVideoX-2B [84], CogVideoX-5B [84], HunyuanVideo [49], and CogVideoX-2B-I2V [84]). For CogVideoX-2B [84], each video has a resolution of $480 \times 720$ and consists of 49 frames. Human annotators carefully select the final prompt-generated video pairs based on motion consistency and overall video fidelity, resulting in the top-50 pairs for each dataset.

**Generating Pseudo Ground-Truth.**   To evaluate temporal matching, our dataset includes pseudo ground-truth trajectories, as no ground-truth trajectories exist for synthesized videos. We generate these using the off-the-shelf point tracking method CoTracker [44]. Specifically, we first define the starting points of the trajectories in the first frame. For the object dataset, which focuses on object dynamics, we segment the centered object using SAM [48] and sample grid points at $1/20$ of the video's spatial resolution. For the scene dataset, where we aim to capture overall camera movement, we uniformly sample a $10 \times 10$ grid across the entire frame. Using CoTracker, we obtain per-point trajectories along with their visibility. For our analysis, we consider only points that CoTracker estimates as visible.

# B  DiffTrack on Other Video DiTs

DiffTrack is compatible with any off-the-shelf video DiT architecture. We extend our analysis to additional DiT-based video models, including CogVideoX-5B [84], HunyuanVideo [49] and CogVideoX-2B-I2V [84].

**DiffTrack on CogVideoX-5B.**  In Fig. A.1, we present an analysis of CogVideoX-5B using Diff-Track. The model consists of 42 transformer layers, and we use 50 sampling timesteps for this study. Our observations reveal that 1) query-key matching achieves higher matching accuracy compared to intermediate feature matching (Fig. A.1(a)), 2) a small number of layers dominate temporal matching (Fig. A.1(b)), and 3) temporal matching improves as noise levels decrease, with a slight drop at the final stages of denoising (Fig. A.1(c)). These findings are consistent with the analysis presented in Sec. 3.4 for CogVideoX-2B.

**DiffTrack on HunyuanVideo.**  Fig. A.2 presents additional analysis of HunyuanVideo using DiffTrack. The model consists of 60 layers, and we use 30 sampling timesteps for this experiment. We observe that: query-key matching outperforms intermediate feature matching (Fig. A.2(a)), a few specific layers play dominant roles in temporal correspondence (Fig. A.2(b)), and temporal matching improves as diffusion noise decreases (Fig. A.2(c)).

**DiffTrack on CogVideoX-5B-I2V.**  In Fig. A.3, we present further analysis of CogVideoX-5B-I2V [84]. We observe that query–key matching outperforms intermediate feature matching (Fig. A.3(a)), a small number of layers dominate temporal correspondence (Fig. A.3(b)), and temporal matching improves over diffusion timesteps (Fig. A.3(c)).

In Tab. A.1, we further evaluate CogVideoX-5B-I2V on point tracking accuracy using the DAVIS dataset, and observe that it outperforms SVD, a U-Net–based model. However, CogVideoX-5B-I2V underperforms compared to its T2V counterpart, CogVideoX-5B. We attribute this to the image-to-video fine-tuning objective, which primarily focuses on preserving the appearance of the first frame. This tends to produce more static

| Backbone | $< \delta^0$ | $< \delta^2$ | $< \delta^4$ | $< \delta^x_{avg}$ |
|---|---|---|---|---|
| SVD [6] | 3.6 | 34.1 | 71.4 | 35.9 |
| DiffTrack (CogVideoX-5B-I2V [84]) | 3.9 | 38.2 | 69.0 | 36.8 |
| DiffTrack (CogVideoX-5B [84]) | **5.2** | **50.7** | **84.3** | **46.9** |

Table A.1: **Quantitative comparison on the DAVIS dataset [19].**

videos and weakens the temporal correspondence required for generating dynamic motion, as also discussed in [17].

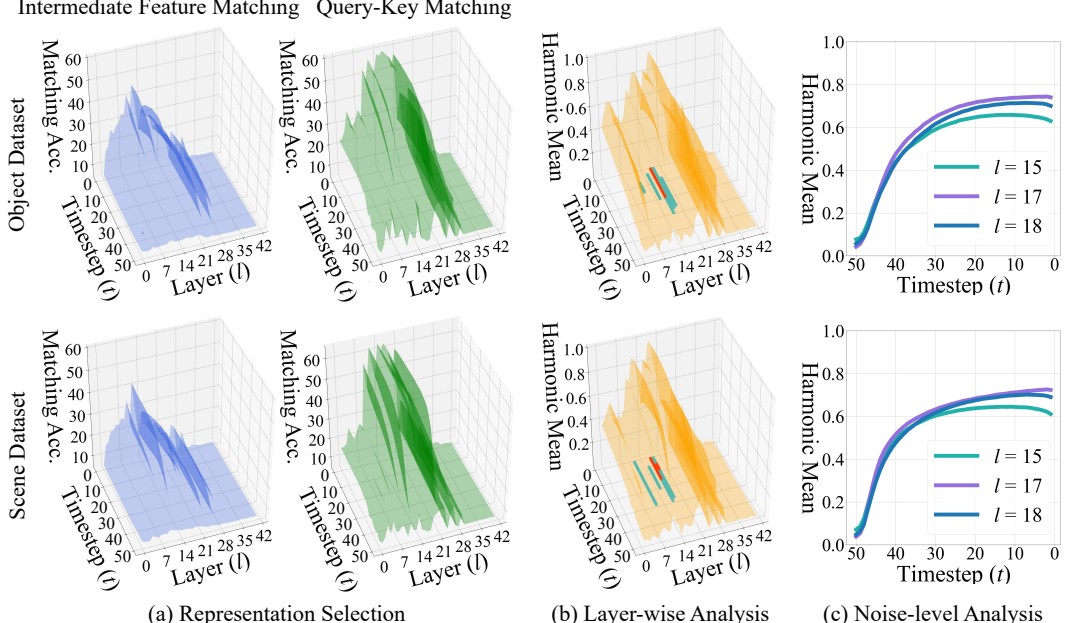

Figure A.1: **Analysis of temporal matching in CogVideoX-5B [84].** (a) *Representation selection*: Query-key matching achieves higher accuracy than intermediate feature matching. (b) *Layer-wise analysis*: Temporal correspondence is primarily governed by a limited set of layers. (c) *Noise-level analysis*: Temporal matching improves as noise decreases but slightly degrades near the final steps.

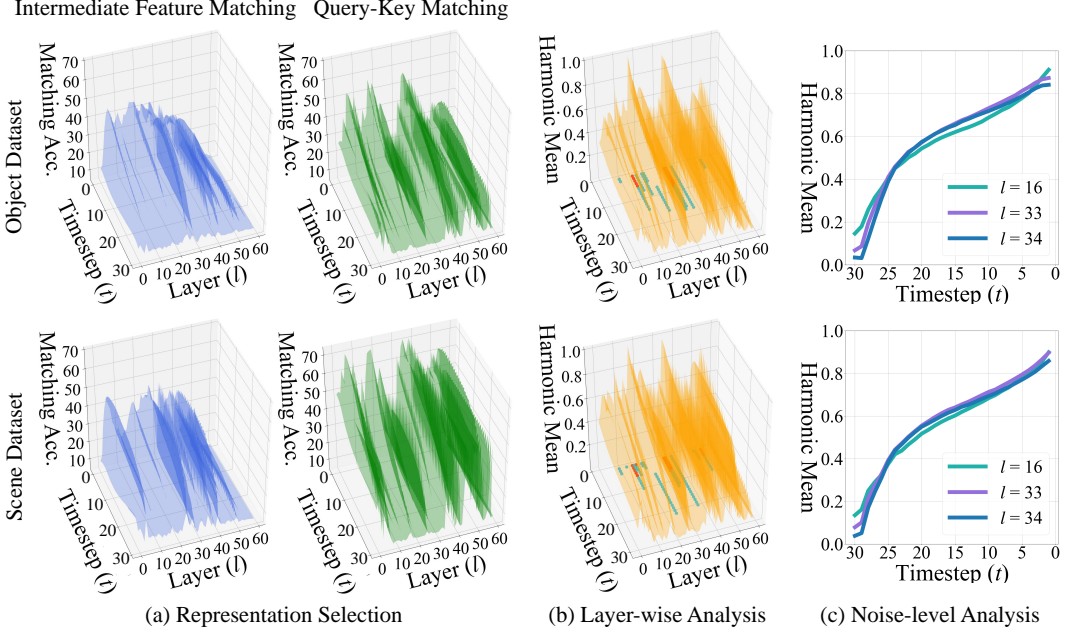

Figure A.2: **Analysis of temporal matching in HunyuanVideo [49].** (a) *Representation selection*: Query-key matching achieves higher accuracy than intermediate feature matching. (b) *Layer-wise analysis*: Temporal correspondence is primarily governed by a limited set of layers. (c) *Noise-level analysis*: Temporal matching improves as noise decreases.

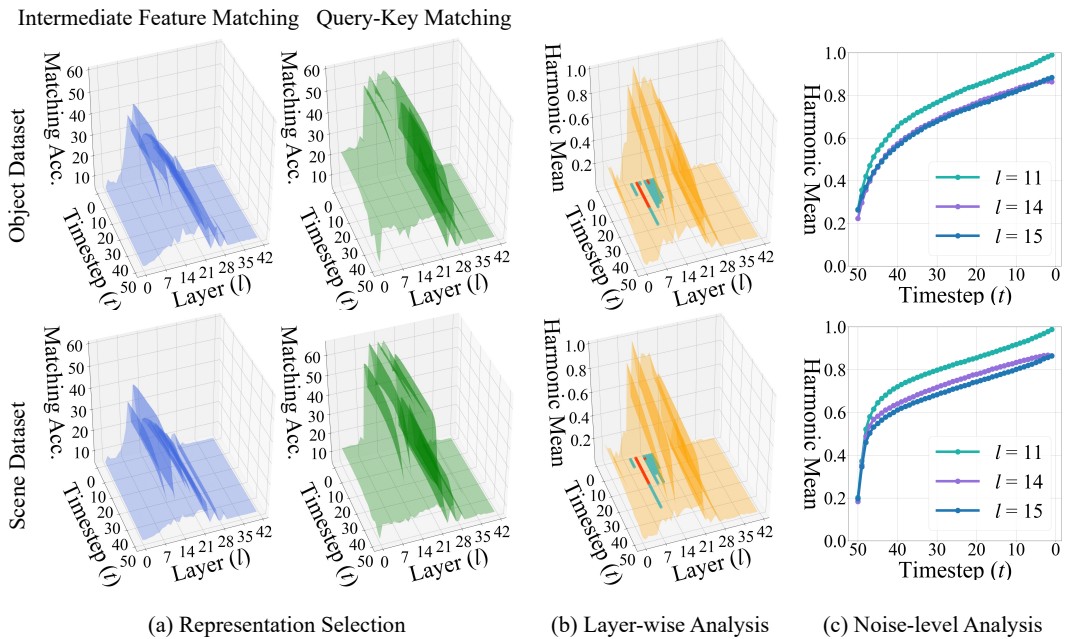

Figure A.3: **Analysis of temporal matching in CogVideoX-5B-I2V [84].** (a) *Representation selection*: Query-key matching achieves higher accuracy than intermediate feature matching. (b) *Layer-wise analysis*: Temporal correspondence is primarily governed by a limited set of layers. (c) *Noise-level analysis*: Temporal matching improves during denoising steps.

## C  Zero-Shot Point Tracking Details

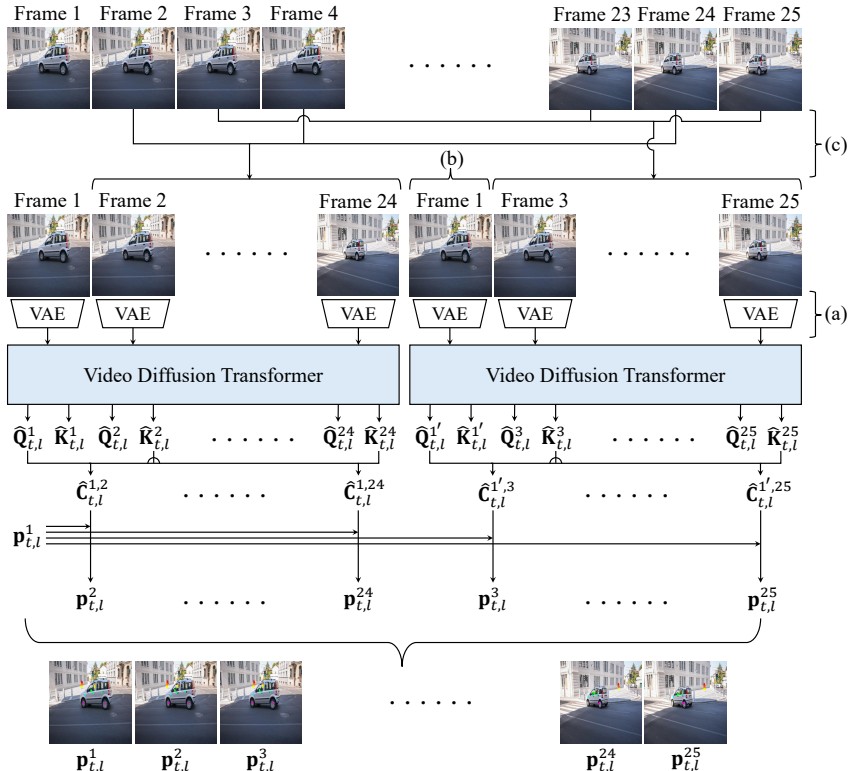

Figure A.4: **Overall architecture of DiffTrack for zero-shot point tracking.** (a) Removing temporal compression improves point accuracy by avoiding interpolation. (b) A global first-frame is inserted into each chunk for direct matching. (c) Interleaved chunk construction reduces the temporal gap to the first frame.

### C.1  Architectural Details

Figure A.4 illustrates the overall architecture of DiffTrack for zero-shot point tracking on real videos. In this example, a 25-frame video is divided into two chunks for visualization. We further provide a component analysis of this architecture in Sec. F.1.

We find that temporal compression in the 3D VAE limits point accuracy due to the linear interpolation used to reconstruct motion trajectories from latent space. To mitigate this, we set the temporal compression ratio to $q = 1$, establishing a one-to-one mapping between each video frame and its latent (Fig. A.4(a)). This is achieved by passing each frame individually to the pre-trained VAE, which compresses $1 + 4f$ frames to $1 + f$ latents, where we set $f = 0$.

To enable direct temporal matching with the global first frame, we insert it into every chunk (Fig. A.4(b)). To reduce the temporal gap between the global first frame and other frames within each chunk, we construct chunks using interleaved subsequent frames (Fig. A.4(c)). Frames within each chunk are sampled at uniform intervals, determined by dividing the total video length by $f - 1$. Chunks slide with a stride of 1, and the matching costs for overlapping frames across chunks are averaged.

In full 3D attention, each frame latent is projected into $\hat{\mathbf{Q}}_{t,l}^i$ and $\hat{\mathbf{K}}_{t,l}^i$, where $i$ denotes the frame index. With slight abuse of notation, we represent the projections of the inserted global first-frame latent in the second chunk as $\hat{\mathbf{Q}}_{t,l}^{1'}$ and $\hat{\mathbf{K}}_{t,l}^{1'}$. For each chunk, we compute the matching cost between the first-frame query and the key of the $j$-th frame (*cf.* Eq. (5)). To enhance performance, following [3, 13, 14, 33–37], we employ a bidirectional matching cost that additionally incorporates the transposed inverse cost between the query of the $j$-th frame and the key of the first frame. This results in $\hat{\mathbf{C}}_{t,l}^{1,j}$ for the first chunk and $\hat{\mathbf{C}}_{t,l}^{1',j}$ for the second chunk. Next, we apply an argmax operation

(*cf.* Eq. (6)) to obtain the matched correspondence points: the starting point $\mathbf{p}_{t,l}^1$ and its match $\mathbf{p}_{t,l}^j$ in the $j$-th frame. By concatenating $\mathbf{p}_{t,l}^j$ across frames, we obtain the full motion trajectory.

## C.2   Implementation Details

**Zero-shot Point Tracking.**   For zero-shot tracking evaluation [5], we used the most significant layer and timestep, identified by matching accuracy in Sec. 3.4 and Sec. B: $l = 17$, $t = 1$ for CogVideoX-2B, $l = 16$, $t = 1$ for for CogVideoX-5B, and $l = 16$, $t = 1$ for HunyuanVideo. All experiments were conducted on an A6000 GPU.

We evaluate zero-shot tracking on two real-video datasets with precisely annotated tracks: TAP-Vid-DAVIS [19] and TAP-Vid-Kinetics [19]. DAVIS includes 30 videos with diverse object motions and appearance variations, while Kinetics comprises 1,189 in-the-wild videos featuring rapid scene transitions and motion blur.

Following prior work [16, 43, 44], we evaluate point accuracy at $256 \times 256$ resolution. For CogVideoX-2B, we resize videos to $256 \times 256$ and then upsample to the training resolution of $480 \times 720$. The resulting feature descriptors have a spatial resolution of $30 \times 45$, as the 3D VAE decompresses spatial size to $1/16$.

**Human Evaluation.**   We also provide an example of human evaluation in Fig. A.16.

## C.3   Comparison

We demonstrate the effectiveness of DiffTrack in zero-shot point tracking [5] by comparing it with a diverse set of vision foundation models trained on single images and self-supervised models trained on videos or two-view images. Below, we detail the models included in our comparison. For fair evaluation, following [5], we resize inputs to produce feature maps of size $30 \times 45$, except for ZeroCo [3].

**Vision Foundation Models.**   We evaluate DiffTrack against vision foundation models, including DINO, DINOv2, DINOv2-Reg, and DIFT (SD1.5, SD2.1).

DINO [10] is a self-supervised vision transformer that learns localized features of salient objects. DINOv2 [60] improves upon DINO by leveraging a larger dataset and optimized training. DINOv2-Reg [18] introduces register tokens to further reduce attention artifacts. We use ViT-B/16 for DINO and ViT-B/14 for both DINOv2 and DINOv2-Reg.

Stable Diffusion 1.5 (SD1.5) and 2.1 (SD2.1) [68] are U-Net-based text-to-image diffusion models. Following DIFT [75], we extract features from the third upsampling block to compute point correspondence.

**Self-Supervised Models.**   We further evaluate DiffTrack using self-supervised video models, including SMTC, CRW, Spa-then-Temp, VFS, and SVD, as well as a self-supervised model trained on two-view images, ZeroCo. All of these models are trained solely on videos or two-view images without any labels.

SMTC [65] proposes a self-supervised video model that improves semantic and temporal consistency by training the architecture in a teacher-student manner. We use SMTC with ViT-S/16 for our comparison.

Contrastive Random Walk (CRW) [40] trains ResNet-18 to learn temporally consistent feature representations through cycle-consistency, maximizing the likelihood of returning to the initial points when walking through palindromic video sequences.

Spa-then-Temp [52] combines spatial and temporal self-supervised learning by first leveraging contrastive learning for spatial features and then enhancing these features through hierarchical frame reconstruction and local correlation distillation. We use Spa-then-Temp based on ResNet-50 for comparison.

Video Frame-level Similarity (VFS) [83] compares frame-level features from the same video as positive pairs, while frames from different videos serve as negative pairs. We use VFS with ResNet-50 for comparison.

Stable Video Diffusion (SVD) [6] is a U-Net-based text-to-video generative model extended from SD 2.1 by incorporating additional temporal layers. Following [41], we use features from the upsampler

layer of the third decoder block to calculate point accuracy. We empirically observe that query-key matching achieves higher point accuracy than feature matching; therefore, we adopt query-key matching in the third decoder block of SVD.

ZeroCo [3] demonstrates query-key matching in the cross-attention map within the self-supervised cross-view completion model CroCo [3], capturing geometric correspondence more effectively than other correlation maps from the encoder or decoder. For comparison, we use an input size of $224 \times 224$ with a feature size of $14 \times 14$, as we empirically found that this trained resolution yields the best point accuracy compared to other input resolutions with higher feature sizes.

# D   Cross-Attention Guidance Details

**Architectural Details.**   In Fig. 9, we present the overall architecture of Cross-Attention Guidance (CAG). Inspired by PAG [2], which enhances image fidelity by transforming selected self-attention maps in the diffusion U-Net into identity matrices, we extend this idea to the video DiT architecture.

In PAG, the identity matrices are created by multiplying a diagonal mask into the attention map before the softmax operation—where diagonal elements are set to 0 and off-diagonal elements to $-\infty$. After softmax, this yields an identity matrix (diagonal values of 1, others 0), allowing values to pass through unchanged.

A naive extension to video assigns $-\infty$ to cross-frame positions and 0 elsewhere before the softmax operation. However, this undesirably suppresses the scale of self-frame and text-frame attention values. To address this, we instead zero out only the cross-frame attention values after softmax in $\mathbf{A}_{t,l}$, producing modified attention maps $\hat{\mathbf{A}}_{t,l}$ that preserve all other interactions.

**Implementation Details.**   For CAG, we used the top-3 dominant layers $l = 13, 17, 21$ for CogVideoX-2B and $l = 15, 17, 18$ for CogVideoX-5B, as identified by harmonic mean in Sec. 3.4 and Fig. A.1. Following [2], we applied the guidance at all sampling timesteps.

**Evaluation Details.**   We evaluate CAG against its baselines, CogVideoX-2B and CogVideoX-5B, on VBench [38]. We used the prompt suite provided by VBench for each evaluation dimension.

To assess temporal quality, we report three metrics: Subject Consistency, Background Consistency, and Dynamic Degree. Subject Consistency measures whether the appearance of the main subject (*e.g.* a person or an object) remains consistent across frames, computed using DINO feature similarity. Background Consistency assesses the temporal coherence of background scenes, measured by CLIP feature similarity across frames. While a completely static video can achieve high scores on the aforementioned temporal quality metrics, it is essential to also evaluate the presence and magnitude of motion. To this end, we calculate Dynamic Degree, which quantifies motion dynamics using optical flow computed by RAFT [76].

To evaluate frame-wise quality, we report Imaging Quality, which detects frame-wise distortions (*e.g.*, blur, noise), computed using the MUSIQ [45] image quality prediction model trained on the SPAQ [22] dataset.

# E   Additional Analysis

In this section, we present additional analysis of CogVideoX-2B [84] using our framework, DiffTrack. To calculate point accuracy (PCK), we use a predefined error threshold $\delta^3$ for all analyses presented in the main paper and appendix.

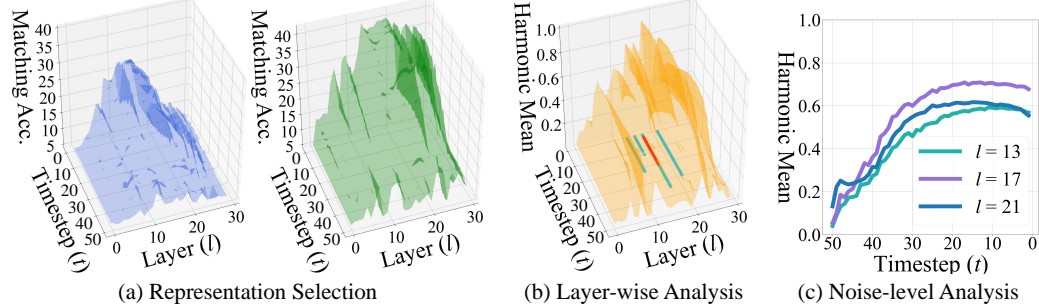

(a) Representation Selection    (b) Layer-wise Analysis    (c) Noise-level Analysis

Figure A.5: **Analysis of temporal matching in CogVideoX-2B [84] with DAVIS [19] dataset.** (a) *Representation selection*: Query-key matching achieves higher accuracy than intermediate feature matching. (b) *Layer-wise analysis*: Temporal correspondence is primarily governed by a limited set of layers. (c) *Noise-level analysis*: Temporal matching improves throughout the denoising process and slightly degrades near the end.

**Analysis with Real Videos.**    In Fig. A.5, we analyze CogVideoX-2B using real videos from the DAVIS dataset [19]. As discussed in Sec. 3.4, reconstructing real videos often introduces inversion errors [71] due to challenges in finding accurate text prompts or discrepancies between real videos and the training distribution.

To address this, instead of applying diffusion inversion, we add Gaussian noise to the frame latents at each timestep $t$. Specifically, we truncate all DAVIS videos to 49 frames, excluding those shorter than 49 frames. At each timestep $t$, we add Gaussian noise corresponding to $t$ to the frame latent, pass it through the diffusion transformer, and extract feature descriptors for analysis. Notably, this approach is consistent with prior work [75, 86], which identifies two-frame correspondences in U-Net-based image diffusion models.

The analysis with DAVIS in Fig. A.5 shows a similar trend to that observed in Fig. 4 using a curated synthetic dataset. Specifically, query-key matching consistently outperforms intermediate feature matching. Additionally, a few layers predominantly contribute to temporal matching, and these dominant layers align with those identified in Fig. 4. Temporal matching strengthens during the denoising process, with slight drops near the end of the timesteps.

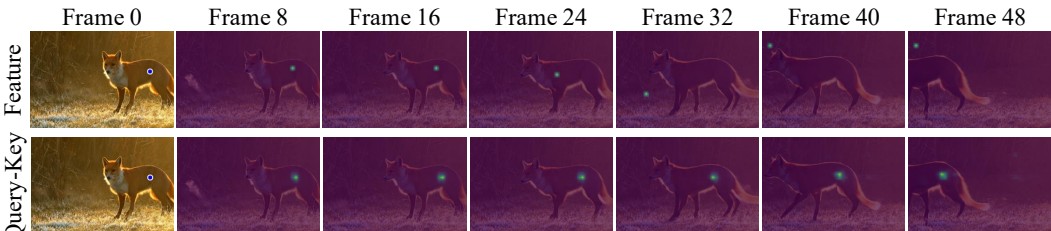

Figure A.6: **Attention visualization comparison between intermediate feature matching and query-key matching.**

**Feature Matching vs. Query-Key Matching.**    Fig. A.6 compares attention maps across timesteps, derived from intermediate feature matching and query-key matching. We observe that query-key matching successfully tracks physically matched points, whereas intermediate feature matching often fails. This further supports the observation in Fig. 4(a) that query-key matching yields better temporal correspondence.

This finding is consistent with prior works [3, 57], which demonstrate that query-key similarities include geometric cues crucial for accurate matching, whereas value warps the visual appearance based on these similarities.

**PCA Visualization.** Fig. A.7 visualizes the Principal Component Analysis (PCA) of queries, keys, and values across video frames. Queries and keys exhibit stronger structural cues, with similar distributions for nearby pixels, enabling more effective geometric matching. In contrast, values contain high-frequency noisy appearance features, potentially diluting structural cues for correspondence.

This analysis aligns with [3, 57], which show that query-key similarities capture structural information crucial for accurate geometric matching, while values encode semantic appearance information, further warped by the query-key similarities.

**Attention Visualization.** Fig. A.17 presents cross-frame attention maps between the first frame and subsequent frames. In the top-ranked layers based on harmonic mean

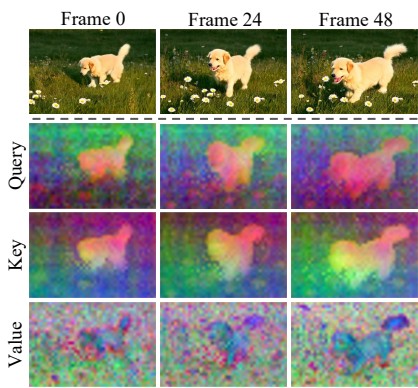

Figure A.7: **PCA visualization of queries, keys and values.**

scores (Fig. A.17(a)), the attention is sharp and accurately localized at the matched point. In contrast, the bottom three layers with the lowest harmonic scores (Fig. A.17(b)) exhibit diffuse and scattered attention patterns. This supports our analysis in Fig. 4(b), which shows that only a few layers contribute significantly to temporal matching.

Fig. A.18 illustrates how cross-frame attention evolves throughout the denoising process, revealing that attention becomes progressively sharper at later timesteps. This observation supports our analysis in Fig. 4(c), which shows that temporal matching improves as denoising progresses.

Fig. A.19 displays the text-to-frame attention maps for the query word "shark" across timesteps. As discussed in Fig. 5, the attention evolves from coarse to fine as noise levels decrease, highlighting that text primarily determines the global semantic layout during the early stages of denoising.

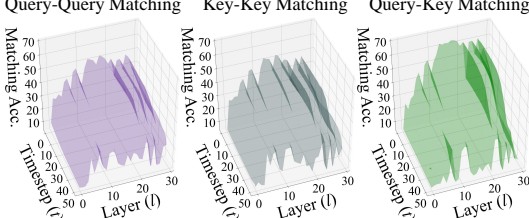

Figure A.8: **Comparison of query-query, key-key, and query-key matching.**

**Query-Query vs. Key-Key vs. Query-Key Matching.** Fig. A.8 shows the matching accuracy of query-query, key-key, and query-key interactions on the object dataset. Query-key matching achieves the highest accuracy, indicating that query-key interactions within full 3D attention inherently learn temporal matching.

# F    Ablation Study

## F.1    Zero-Shot Point Tracking

**Component Analysis.**    Tab. A.2 ab-
lates the effects of temporal com-
pression and long-term sequence han-
dling in zero-shot point tracking per-
formance on DAVIS. We evaluate four
configurations: (I) the baseline, which
employs temporal compression in the
3D VAE and sequential chunking; (II)

| | Component | $< \delta^0$ | $< \delta^2$ | $< \delta^4$ | $< \delta^x_{\text{avg}}$ |
|---|---|---|---|---|---|
| (I) | Baseline | 2.6 | 21.9 | 60.2 | 26.9 |
| (II) | (I) + w/o temporal compression | 2.7 | 31.6 | 64.5 | 32.3 |
| (III) | (II) + w/ first-frame insertion | 4.7 | 47.6 | 70.5 | 44.6 |
| (IV) | (III) + w/ interleaved frames | **4.8** | **49.2** | **84.3** | **46.3** |

Table A.2: **Ablation study:** analyzing the impact of temporal
compression and long-term handling.

the baseline without temporal compression ($q = 1$), ensuring a direct one-to-one mapping between
frames and frame latents; (III) an approach that inserts the first frame into every chunk, enabling
explicit interaction with the first frame in all chunks; and (IV) our final method, which further applies
interleaved frame construction to better handle long-term context. Compared to (I), (II) improves per-
formance by eliminating interpolation errors introduced by temporal compression. Compared to (II),
(III) shows a significant improvement, demonstrating the importance of direct temporal interaction
with the global first frame within each chunk. Additionally, (IV) further enhances performance by
interleaving frames within each chunk, effectively reducing the frame interval between the first and
other frames.

**Impact of Feature Selection.**    Tab. A.3 highlights the
effectiveness of feature selection by comparing matching
accuracy on the object dataset using three strategies: (I)
averaging matching costs over all timesteps at the most
dominant layer ($l = 17$), (II) averaging over all layers at
the most dominant timestep based on matching accuracy
($t = 1$), and (III) using matching costs from the most
dominant layer and timestep ($l = 17, t = 1$). Selecting

| | Layer ($l$) | Timestep ($t$) | $< \delta^3$ |
|---|---|---|---|
| (I) | $l = 17$ | $t \in [1, 50]$ | 15.34 |
| (II) | $l \in [0, 29]$ | $t = 1$ | 16.69 |
| (III) | $l = 17$ | $t = 1$ | **63.50** |

Table A.3: **Ablation study,** analyzing the
impact of feature selection.

optimal features significantly improves accuracy, emphasizing the value of our analysis and the
importance of feature selection for reliable temporal matching.

**Multi-Feature Fusion for Zero-Shot Point
Tracking.**    Prior works [31, 42, 56, 57, 75, 86]
on intermediate diffusion features suggest that
fusing multiple timesteps and layers improves
semantic correspondence by leveraging the hi-
erarchical structure of diffusion representations.
Motivated by this, we investigate whether such
fusion also benefits temporal matching.

| Layer ($l$) | Timestep ($t$) | $< \delta^0$ | $< \delta^2$ | $< \delta^4$ | $< \delta^x_{\text{avg}}$ |
|---|---|---|---|---|---|
| $l = 17$ | $t = 1$ | 4.8 | 49.2 | 84.3 | 46.3 |
| $l = 13, 17, 18$ | $t = 1$ | 4.9 | 48.6 | 84.2 | 46.0 |
| $l = 17$ | $t = 1, 2, 3$ | 4.9 | 49.3 | 85.0 | 46.6 |

Table A.4: **Ablation study,** analyzing fusing features
across multiple timesteps and layers.

Tab. A.4 summarizes results on the DAVIS
dataset [64]. In contrast to prior findings in seman-
tic correspondence, we observe that fusing across
timesteps and layers has a negligible impact on
point accuracy. This discrepancy arises because
temporal matching requires precise, pixel-level ge-
ometric alignment to track the same physical point
across frames, whereas semantic correspondence
benefits from multi-scale contextual cues to match
similar regions.

PCA visualizations in Fig. A.9 support this finding.
In Fig. A.9(a) and (b), query-key features from dif-
ferent timesteps exhibit high similarity, indicating
limited benefit from temporal fusion. In contrast,
Fig. A.9(c) and (d) show greater variation across
features from different layers, suggesting that the
tracking performance is primarily driven by a single
dominant layer and that layer fusion introduces noise rather than informative diversity.

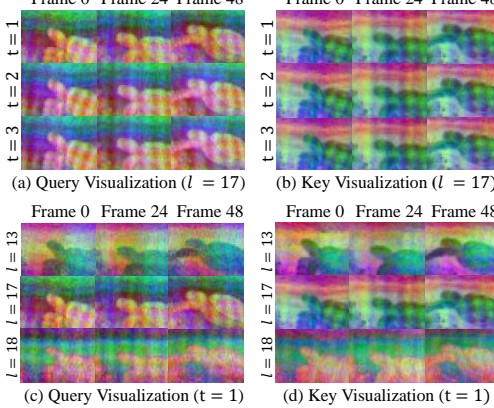

Figure A.9: **PCA visualizations of query-key fea-
tures across timesteps and layers.**

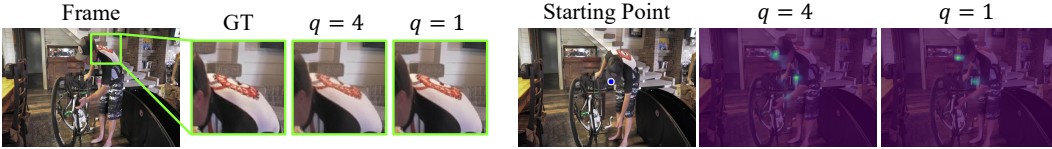

| Frame | GT | $q = 4$ | $q = 1$ | Starting Point | $q = 4$ | $q = 1$ |

(a) Reconstructed Frames by VAE Decoder  (b) Attention Map Visualization

Figure A.10: **Analysis of temporal compression.** (a) Reconstructed frames from the VAE decoder using different temporal compression ratios $q$. (b) Visualization of attention maps under different temporal compression ratios.

**Temporal Compression.** In Fig. A.10, we present decoded frames from the 3D VAE decoder and corresponding attention visualizations for different temporal compression ratios $q$. (a) For $q = 4$, the reconstructed frame is relatively blurred due to the limited expressivity of the compressed representation, while $q = 1$ allows each frame to be reconstructed almost perfectly without compression. (b) The attention visualization for $q = 4$ shows more diffuse attention, while $q = 1$ exhibits sharper, more focused attention. This supports our architectural design in Fig. A.4(a), which contributes to improved point accuracy.

| #Frames | $< \delta^0$ | $< \delta^2$ | $< \delta^4$ | $< \delta^x_{\text{avg}}$ |
|---|---|---|---|---|
| 2 | 4.4 | 43.9 | 77.1 | 41.7 |
| 5 | 4.5 | 46.3 | 82.2 | 44.3 |
| 8 | 4.7 | 47.9 | 83.8 | 45.5 |
| 11 | 4.8 | 48.5 | 84.3 | 46.1 |
| 13 | **4.8** | **49.2** | **84.3** | **46.3** |

| Starting Point | # of Frames = 2 | # of Frames = 13 |

Figure A.11: **Ablation study** on the number of frames per chunk.

Figure A.12: **Attention map for different number of frames.**

**Number of Frames per Chunk.** Fig. A.11 ablates matching accuracy on DAVIS using different numbers of frames per chunk. We observe that as the number of frames increases, matching accuracy consistently improves. This is further supported by Fig. A.12, which visualizes attention maps across different chunk sizes, showing that longer chunks lead to sharper attention. This suggests that multi-frame interaction enhances temporal correspondence, highlighting the role of cross-frame attention over multiple frames within video DiTs, in contrast to image-based models [6, 86].

### F.2 Motion-Enhanced Video Generation

**Layer Selection.** Tab. A.5 compares the least top-3 and most top-3 dominant layers for temporal matching ($l = 30, 31, 32$ vs. $l = 13, 17, 18$) of CogVideoX-5B in motion-enhanced video generation. We observe that guiding with the most dominant layers ($l = 13, 17, 18$) consistently improves all evaluation metrics compared to the baseline, whereas

| Method | Subject Consistency | Background Consistency | Dynamic Degree | Imaging Quality |
|---|---|---|---|---|
| CogVideoX-5B | 0.9158 | 0.9590 | 0.6667 | 0.5531 |
| CogVideoX-5B + **CAG** ($l = 30, 31, 32$) | 0.9147 | 0.9580 | **0.7059** | 0.5683 |
| CogVideoX-5B + **CAG** ($l = 13, 17, 18$) | **0.9283** | **0.9644** | 0.6863 | **0.6051** |

Table A.5: **Quantitative comparison of CAG applied at different layers.**

guiding with the least dominant layers ($l = 30, 31, 32$) results in only marginal gains in Image Quality and even lower performance in Subject Consistency and Background Consistency, under the same guidance scale. These findings further support our analysis that specific layers play a critical role in enhancing temporal consistency during generation.

**Memory and Time Consumption.** Tab. A.6 compares inference time and memory usage per denoising timestep across three settings: the baseline, baseline with Classifier-Free Guidance (CFG), and baseline with CFG and Cross-Attention

| Method | Memory (GB) | Time (s) |
|---|---|---|
| Baseline (CogVideoX-2B) | 2.75 | 1.31 |
| Baseline + **CFG** | 12.75 | 2.60 (198%) |
| Baseline + **CFG** + **CAG** | 12.84 | 4.59 (177%) |

Table A.6: **Memory and Time Consumption.**

Guidance (CAG). While inference runtime increases due to the increase of the number of function evaluations, CAG's GPU memory footprint remains very similar to that of CFG since CAG simply applies an attention mask to a few dominant layers.

This stands in contrast to prior works [11, 41], which introduce additional modules and require training for motion enhancement. In comparison, our method is entirely training-free and operates within the model itself via a novel guidance mechanism.

| Backbone | $< \delta^0$ | $< \delta^1$ | $< \delta^2$ | $< \delta^3$ | $< \delta^4$ | $< \delta^x_{\text{avg}}$ |
|---|---|---|---|---|---|---|
| DIFT (SD1.5) [75] | 3.5 | 13.0 | 39.3 | 63.1 | 72.2 | 38.2 |
| DIFT (SD2.1) [75] | 3.6 | 13.3 | 40.1 | 65.8 | 75.7 | 39.7 |
| Diffusion Hyperfeatures (SD1.5) [55] | 3.0 | 12.9 | 33.1 | 58.4 | 74.5 | 36.4 |
| Diffusion Hyperfeatures (SD2.1) [55] | 3.7 | 15.5 | 38.8 | 64.5 | 78.9 | 40.3 |
| CleanDIFT (SD1.5) [72] | 3.1 | 13.7 | 36.5 | 64.9 | 79.6 | 39.6 |
| CleanDIFT (SD2.1) [72] | 3.7 | 16.0 | 41.8 | 68.6 | 81.9 | 42.4 |
| TLR (SD1.5 + DINOv2 (ViT-B/14)) [85] | 4.3 | 17.7 | 44.6 | 69.8 | 82.0s | 43.7 |
| DiffTrack (CogVideoX-2B [84]) | 4.8 | 19.4 | 49.2 | 73.6 | **84.3** | 46.3 |
| DiffTrack (CogVideoX-5B [84]) | **5.2** | **20.5** | **50.7** | **73.9** | **84.3** | **46.9** |

Table A.7: **Quantitative comparison on the DAVIS dataset [19].**

# G  Additional Qualitative Results

Additional qualitative results for zero-shot point tracking and motion-enhanced video generation are provided in Fig. A.20 and Fig. A.21.

# H  Additional Quantitative Results

Additional quantitative results for zero-shot point tracking on the DAVIS dataset are provided in Tab. A.7.

# I  Related Works

**Video Diffusion Models.** Early approaches [4, 6, 7, 12, 27, 47, 51, 70, 82] were primarily based on U-Net [32, 62, 68], often achieved by inflating pre-trained image diffusion models [32, 62, 68], typically with separate spatial and temporal attention mechanisms. Although efficient, this separation restricts direct frame-to-frame spatial and temporal interactions, leading to temporal inconsistency or a lack of large motions in generated videos. Sora [54] has demonstrated the effectiveness of Diffusion Transformers (DiTs) [21] in increasing scalability and improving temporal coherence. Subsequent works [25, 28, 50, 63, 69, 84, 88], such as CogVideoX [84], MovieGen [63], Mochi1 [25], and LTX-Video [28], have adopted the DiT architecture, achieving unprecedented performance. Unlike U-Net-based methods, DiTs employ full 3D attention, enabling cross-frame information sharing between frame latents as well as with text embeddings. This explicit cross-frame attention improves temporal coherence [84]. Despite these advances, how video DiTs capture temporal correspondence during video generation remains unexplored.

**Representation Analysis in Video Diffusion Models.** Recent works [41, 81] explore internal representations in video diffusion models [6, 26, 27] for controlled video generation and improved motion consistency but do not analyze temporal correspondence and are based on U-Nets [32, 62, 68], which are known to struggle with large motion. Another work [8] explores attention control in video DiT [84] for subject consistency in long video generation but focuses solely on text-to-video attention, overlooking temporal matching between frames.

**Exploring Correspondence in Diffusion Models.** Numerous studies [31, 42, 56, 57, 59, 75, 86] have explored intermediate features from pre-trained image diffusion models [32, 62, 68] for correspondence [14, 58, 77, 78]. However, their analyses mainly focus on two-frame correspondence, as image diffusion models are not designed for temporal correspondence in video sequences.

**Temporal Correspondence.** TAP-Vid [19] formulates temporal correspondence in video sequences as point tracking, aiming to estimate the motion of physical points across frames. PIPs [29] iteratively refine estimated trajectories within temporal windows, while TAPIR [20] incorporates depthwise convolutions and enhances initialization. CoTracker [44] jointly tracks near-dense trajectories using spatial correlations. These methods often rely on training with synthetic datasets [19], as annotating real-world data is highly challenging. To address this, a recent study [5] explored zero-shot tracking with visual foundation models [10, 18, 48, 60, 66, 68], showing promising results. However, these analyses are limited to single-image models such as Stable Diffusion (SD) [68] and DINOv2 [60], which process a single frame and thus lack temporal awareness.

A vibrant honeybee, its wings shimmering in the sunlight, delicately lands on a blooming lavender flower, its tiny legs brushing against the soft ...
A vibrant forest scene unfolds as the camera gracefully moves through the lush canopy, revealing intricate bird nests nestled among the branches ...
A close-up view reveals a snail with a glistening, spiraled shell, slowly traversing a lush, dew-kissed leaf. The camera captures the intricate ...
A close-up view reveals a brown caterpillar with intricate patterns along its segmented body, slowly inching across a vibrant green leaf. ...
A majestic eagle perches on a sturdy tree branch, its sharp eyes scanning the vast landscape below. The bird's powerful talons grip the rough ...
A majestic wolf stands in a snowy forest, its thick fur a blend of grays and whites, glistening under the soft winter sunlight. The camera ...
A majestic white fox, with its pristine fur glistening under the soft glow of the moonlight, perches gracefully atop a rugged, moss-covered rock. ...
A meticulously crafted horse figurine stands majestically on a polished wooden surface, its glossy finish reflecting the ambient light. The camera ...
A close-up reveals a vibrant, shimmering fish caught in a woven net, its scales glistening with iridescent hues of silver and blue under the ...
A majestic lion stands regally on a vast expanse of golden wild grass, its mane flowing in the gentle breeze under the warm, golden glow ...
In a serene meadow bathed in the golden light of dawn, a graceful deer with a sleek, tawny coat grazes peacefully amidst a sea of ...
A majestic herd of elephants roams the vast savanna, their massive forms silhouetted against the golden hues of a setting sun. The leader, a wise ...
A detailed close-up captures a fly perched on a vibrant green leaf, its iridescent wings shimmering with hues of blue and green under the soft ...
A majestic cheetah reclines gracefully on a sun-dappled savannah, its sleek, spotted coat blending seamlessly with the golden grass. ...
A close-up shot captures a kangaroo in its natural habitat, its fur a rich blend of earthy browns and grays, as it gently scratches its ...
A majestic great blue heron stands gracefully at the edge of a tranquil lakeside, its long neck elegantly curved, and its striking blue-gray ...
A solitary seagull, with pristine white feathers and a hint of gray on its wings, gracefully strolls along the sandy shore, its slender legs ...
An American crocodile basks on a sunlit riverbank, its rough, scaly skin glistening under the warm sunlight, showcasing shades of olive and gray. ...
A curious wild rabbit with soft, brown fur and twitching whiskers sits alertly in a lush, green meadow, surrounded by vibrant wildflowers and tall ...
A majestic clouded leopard, with its distinctive dusky rosettes and elongated tail, gracefully perches on a sturdy tree branch high above the ...
An African penguin waddles gracefully across a sunlit beach, its distinctive black and white plumage contrasting against the golden sand. ...
A natterjack toad, with its distinctive olive-green skin adorned with warts and a striking yellow stripe down its back, rests on a sunlit rock. ...
In a vibrant animation, a majestic whale emerges, crafted entirely from disposable objects like plastic bottles, straws, and bags, each piece ...
A whimsical scene unfolds with intricately crafted paper cutouts, each element meticulously detailed. Two delicate hands, with visible paper ...
A vibrant pink plastic flamingo, perched on a lush green lawn, sways precariously as a gusty wind sweeps across the scene, causing its slender ...
A curious monkey sits atop a weathered stone, surrounded by lush greenery, its fur a mix of earthy browns and grays, blending seamlessly with the ...
In a dimly lit cave, a solitary bat hangs upside down from the rocky ceiling, its wings wrapped snugly around its small, furry body. ...
A sleek harbor seal glides gracefully through the crystal-clear waters near the rocky shoreline, its smooth, speckled gray coat shimmering under ...
A majestic great white shark glides gracefully through the crystal-clear ocean waters, its powerful body cutting effortlessly through the gentle ...
A majestic goat with a thick, shaggy coat and impressive curved horns stands proudly atop a rugged rock formation, its silhouette framed against a ...
A vibrant butterfly, with iridescent wings displaying a kaleidoscope of blues, purples, and oranges, delicately perches on a budding flower in a ...
A glossy, iridescent beetle slowly emerges from the golden sand, its shell glistening under the warm sunlight. The grains of sand cascade off its ...
A playful penguin chick, its fluffy gray feathers ruffled by the Antarctic breeze, waddles clumsily across the icy terrain. Each step sends tiny ...
A majestic bald eagle perches atop a jagged cliff, its talons gripping the weathered rock as the wind ruffles its pristine feathers. Its sharp ...
A curious meerkat, standing tall on its hind legs, peers across the sun-scorched sands of the African desert, its sleek fur dusted with fine ...
A majestic elk, its towering antlers crowned with strands of golden moss, stands poised at the edge of a sun-drenched meadow. Its dark eyes gleam ...
A curious red fox, its russet fur illuminated by the golden hues of dawn, stands poised on the edge of a frost-kissed meadow. Delicate puffs ...
A majestic moose, its massive antlers adorned with strands of autumn leaves, stands partially submerged in a tranquil forest pond. Ripples spread ...
A curious sea turtle, its shell dappled with hues of emerald and bronze, drifts gracefully through the clear blue waters of a vibrant coral reef. ...
A majestic gray wolf, its thick fur dusted with snow, stands atop a rugged cliff as the pale light of dawn breaks across the distant ...
A regal stag, its antlers crowned with frost-kissed leaves, stands poised atop a hill bathed in the golden hues of dawn. Its dark eyes gleam ...
A curious red fox, its russet coat vibrant against the snowy backdrop of a winter forest, peers out from behind a frost-covered pine tree. Its ...
A graceful swan, its snowy white feathers shimmering in the golden light of sunset, glides serenely across a glassy lake. Gentle ripples fan out ...
A majestic bison, its massive frame dusted with frost, stands resolutely amid the snow-covered plains of the American wilderness. Warm breath ...
A sleek manta ray, its wings spanning wide as it glides effortlessly through the azure depths of the ocean, moves with fluid grace. Sunlight ...
A vibrant green iguana, its textured scales shimmering in shades of emerald and jade, basks atop a sun-warmed rock beneath the canopy of a ...
A graceful swan surveys its surroundings in the serene lake, with shimmering wings reflecting the golden light of dawn. The scene is filled with ...
A proud stag stands motionless atop a snow-dusted hill, its antlers adorned with delicate strands of frost that glisten in the pale winter ...
A graceful white swan glides serenely across the glassy surface of a moonlit lake, its reflection shimmering with each ripple of the water. ...
A powerful wolf, its thick fur a blend of silver and charcoal, stands poised atop a rocky ledge overlooking a mist-shrouded forest. Its amber eyes ...

Figure A.13: **Evaluation prompts for the object dataset.** Our high-quality text prompts are curated from existing benchmarks and generated by a large language model, followed by human annotation to ensure motion consistency and video fidelity in the generated videos.

A drone gracefully glides over the hauntingly silent, abandoned school building in Pripyat, Ukraine, capturing the eerie beauty of its decaying ...
A charming house front door, adorned with festive Christmas decorations, stands as the centerpiece of a cozy winter scene. The door, painted a ...
In the heart of an ancient temple, a sacred sculpture stands majestically, bathed in the soft glow of flickering candlelight. The intricate ...
A striking low-angle view captures the towering facade of a modern apartment building, its sleek glass windows reflecting the vibrant hues of the ...
A breathtaking panorama reveals an ancient Asian temple complex, nestled amidst lush green hills, with intricately carved stone pagodas and ornate ...
A breathtaking aerial view captures the iconic Stuttgart TV Tower, standing tall amidst a lush, verdant forest, its sleek, modern design ...
A breathtaking view unfolds as the camera pans upward, capturing the sky framed by towering skyscrapers. The buildings, with their sleek glass ...
A winding, unpaved road stretches through a lush, verdant landscape, flanked by towering trees with vibrant green leaves, casting dappled shadows ...
A sleek drone glides over a secluded house nestled amidst lush tropical vegetation, capturing the vibrant greens of towering palm trees and dense ...
In a dense, misty forest, towering trees surround several controlled slash piles, their flames flickering and crackling, casting a warm glow ...
In a misty, moonlit garden, a carved jack-o'-lantern with a mischievous grin sits prominently on a rustic wooden table, surrounded by an array of ...
In a tranquil meadow at dawn, the sun's golden rays pierce through a delicate spider web, intricately woven between two tall blades of grass. ...
In a serene winter forest, delicate snowflakes gently blanket the intricate branches of towering trees, creating a mesmerizing tapestry of white ...
A solitary palm tree stands tall and majestic, its slender trunk reaching skyward, crowned with a lush canopy of vibrant green fronds that sway ...
A breathtaking aerial view reveals a vast, snow-covered landscape, where enormous snow piles create a mesmerizing pattern across the terrain. The ...
A breathtaking aerial view captures the first light of dawn as it spills over majestic mountain peaks, casting long shadows across the rugged ...
A breathtaking panorama unfolds, revealing a majestic mountain range cascading into a tranquil sea, dotted with charming islets. These islets, ...
In the haunting remains of an abandoned house, vibrant green grass and resilient plants weave through cracked floorboards and crumbling walls, ...
An aerial view reveals a stunning Croatian bay, where turquoise waters gently lap against the rugged coastline, dotted with lush greenery and ...
From a breathtaking aerial perspective, the camera sweeps over a bustling cityscape, revealing a stunning array of skyscrapers piercing the sky. ...
A cozy bedroom bathed in soft morning light features a plush, king-sized bed with a tufted headboard, adorned with crisp white linens and a ...
The grand interior of a Jewish synagogue unfolds, showcasing intricate architectural details and a serene atmosphere. The space is adorned with ...
An expansive aerial view reveals the vast interior of a bustling warehouse, where rows of towering shelves are meticulously organized with a ...
In an opulent ballroom, grand chandeliers hang from the ornate ceiling, their crystal prisms casting a kaleidoscope of light across the polished ...
In a dimly lit, abandoned indoor swimming pool, the once vibrant tiles now cracked and faded, echo tales of forgotten laughter and splashes. ...
Inside the grand, decaying halls of an abandoned mansion, vibrant graffiti art covers the cracked, peeling walls, transforming the space into a ...
A cozy living room is transformed into a warm haven, featuring a plush beige sofa adorned with soft, colorful cushions, and a rustic wooden coffee ...
A cozy bedroom features a striking exposed brick wall, adding rustic charm to the space. The room is softly lit by a vintage floor lamp, ...
In a sleek, contemporary home studio, a state-of-the-art digital audio workstation sits at the center, surrounded by dual high-resolution monitors ...
A pristine bathroom bathed in soft, natural light features a sleek, modern design. The centerpiece is a freestanding white bathtub with elegant ...
A winding forest road, flanked by towering trees with lush green foliage, stretches into the distance under a canopy of dappled sunlight. ...
A cozy, sunlit kitchen with rustic wooden cabinets and a large farmhouse sink, where morning light streams through a window adorned with lace ...
A grand, indoor library with towering wooden bookshelves filled with countless books, their spines in various colors and textures, stretches up to ...
A cozy nursery bathed in soft, natural light features pastel-colored walls adorned with whimsical animal murals. A white crib with a mobile of ...
Rising from the icy expanse of Antarctica, a breathtaking ice palace shimmers like a crystalline jewel. Walls of translucent ice refract the ...
With its soaring dome and marble facade, a grand Renaissance cathedral dominates the skyline of Florence. Ornate carvings and frescoed ceilings ...
In the heart of a sun-drenched Spanish town, a historic square comes alive with vibrant, whitewashed buildings with ...
Sunlight floods through large windows into a vibrant artist's studio, illuminating canvases propped against the walls. An easel stands at the ...
Rich mahogany shelves, filled with leather-bound books, line the walls of a vintage library. A crackling fireplace casts flickering light on the ...
A minimalist home gym is equipped with sleek black dumbbells, a stationary bike, and a yoga mat laid out on the polished wooden floor. Large ...
In a bright, cheerful kitchen, the air is filled with the scent of freshly cut flowers and citrus fruits. A vase of sunflowers sits on ...
Raindrops patter gently against the window, creating a soothing backdrop to a cozy living room. A soft sofa, draped with knitted blankets and ...
A vintage study filled with leather-bound books and antique furniture exudes timeless elegance. A mahogany desk, cluttered with handwritten notes ...
A tranquil meditation room bathed in soft natural light offers a space for quiet reflection. Floor cushions in earthy tones are arranged in a ...
In a rustic workshop with exposed wooden beams and stone walls, the air is filled with the scent of freshly cut wood and sawdust. ...
Soft lighting and warm textiles create a haven of comfort in a cozy bedroom. A bed, dressed in linen sheets and a knitted blanket, rests ...
A futuristic skyscraper, with its twisting glass facade, pierces the skyline of a bustling metropolis. Its reflective surface mirrors the clouds ...
A colossal Art Nouveau opera house commands attention with its flowing, organic forms and intricate ironwork. Delicate floral motifs swirl across ...
A majestic Renaissance cathedral stands proudly in the heart of an ancient European city, its marble facade adorned with intricate carvings and ...
A grand Byzantine basilica, with its massive central dome and gilded mosaics, stands as a testament to centuries of architectural mastery. ...

Figure A.14: **Evaluation prompts for the scene dataset.** Our high-quality text prompts are curated from existing benchmarks and generated by a large language model, followed by human annotation to ensure motion consistency and video fidelity in the generated videos.

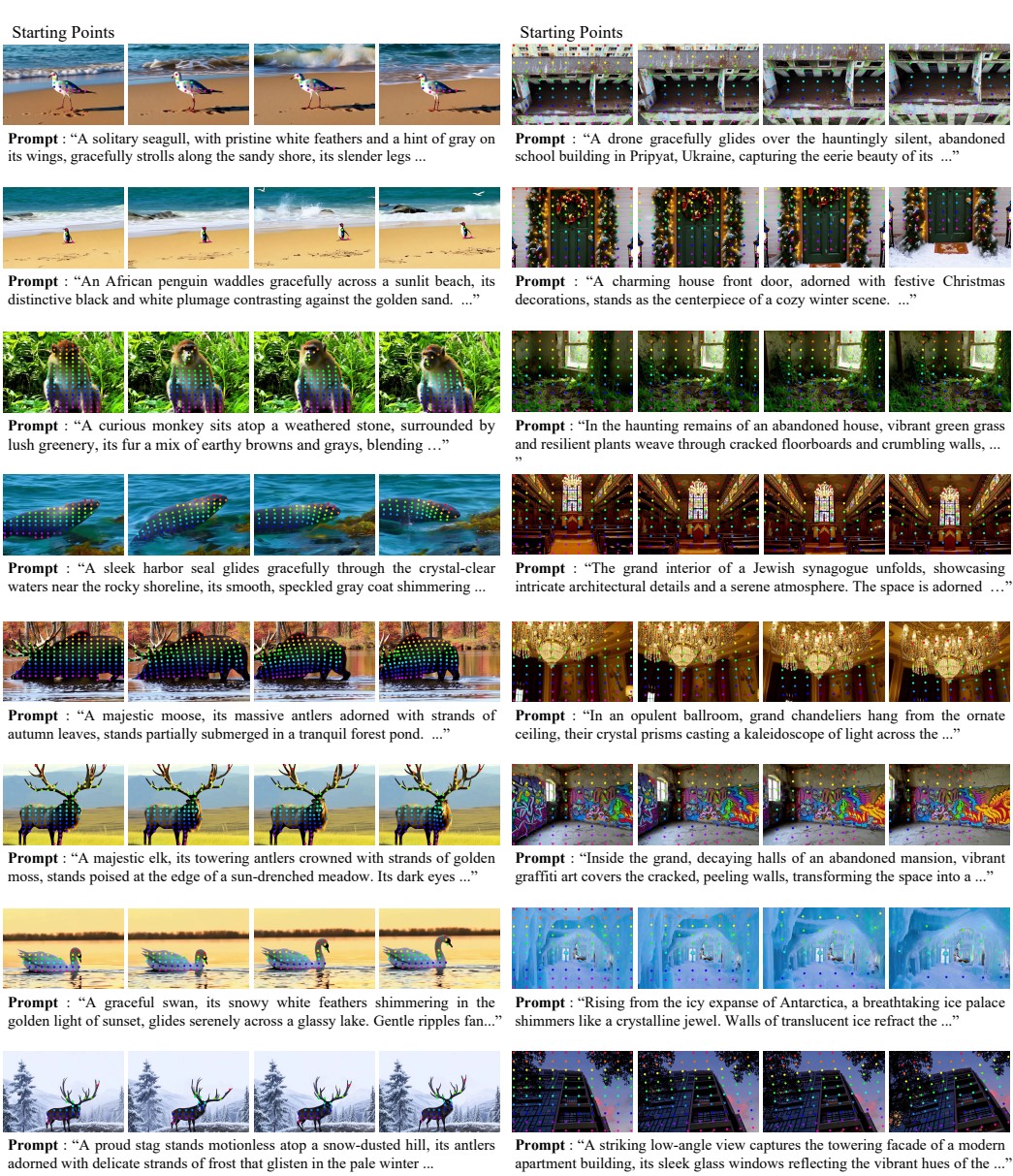

Starting Points

**Prompt** : "A solitary seagull, with pristine white feathers and a hint of gray on its wings, gracefully strolls along the sandy shore, its slender legs ...

**Prompt** : "An African penguin waddles gracefully across a sunlit beach, its distinctive black and white plumage contrasting against the golden sand. ..."

**Prompt** : "A curious monkey sits atop a weathered stone, surrounded by lush greenery, its fur a mix of earthy browns and grays, blending …"

**Prompt** : "A sleek harbor seal glides gracefully through the crystal-clear waters near the rocky shoreline, its smooth, speckled gray coat shimmering ...

**Prompt** : "A majestic moose, its massive antlers adorned with strands of autumn leaves, stands partially submerged in a tranquil forest pond. ..."

**Prompt** : "A majestic elk, its towering antlers crowned with strands of golden moss, stands poised at the edge of a sun-drenched meadow. Its dark eyes ...

**Prompt** : "A graceful swan, its snowy white feathers shimmering in the golden light of sunset, glides serenely across a glassy lake. Gentle ripples fan..."

**Prompt** : "A proud stag stands motionless atop a snow-dusted hill, its antlers adorned with delicate strands of frost that glisten in the pale winter ...

(a) Object Dataset

Starting Points

**Prompt** : "A drone gracefully glides over the hauntingly silent, abandoned school building in Pripyat, Ukraine, capturing the eerie beauty of its ..."

**Prompt** : "A charming house front door, adorned with festive Christmas decorations, stands as the centerpiece of a cozy winter scene. ..."

**Prompt** : "In the haunting remains of an abandoned house, vibrant green grass and resilient plants weave through cracked floorboards and crumbling walls, ..."

**Prompt** : "The grand interior of a Jewish synagogue unfolds, showcasing intricate architectural details and a serene atmosphere. The space is adorned …"

**Prompt** : "In an opulent ballroom, grand chandeliers hang from the ornate ceiling, their crystal prisms casting a kaleidoscope of light across the ..."

**Prompt** : "Inside the grand, decaying halls of an abandoned mansion, vibrant graffiti art covers the cracked, peeling walls, transforming the space into a ..."

**Prompt** : "Rising from the icy expanse of Antarctica, a breathtaking ice palace shimmers like a crystalline jewel. Walls of translucent ice refract the ..."

**Prompt** : "A striking low-angle view captures the towering facade of a modern apartment building, its sleek glass windows reflecting the vibrant hues of the ..."

(b) Scene Dataset

Figure A.15: **Additional examples of our curated dataset.** (a) An object dataset for dynamic object-centric videos and (b) a scene dataset for static scenes with camera motion. The dataset includes predefined starting points in the first frame and their pseudo ground-truth trajectories, obtained using an off-the-shelf tracking method.

The survey comprises 10 sections with a total of 30 questions and will take approximately 5–7 minutes to complete.

In the following pages, you will be presented with pairs of videos and a text prompt.

Please choose the video you consider superior based on the following criteria:

1. *Text Alignment*: The degree to which the video reflects the given text.

2. *Video Quality*: The overall visual quality of the video.

3. *Motion Fidelity*: The naturalness and smoothness of motion in the video.

| Video 1 | Video 2 |

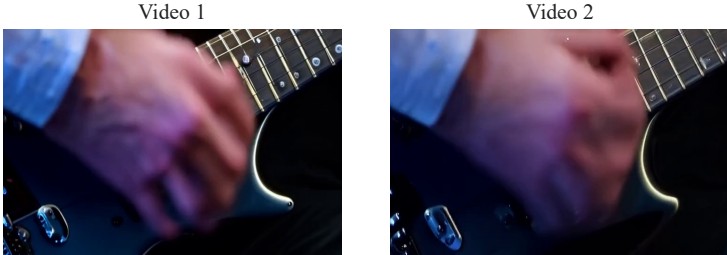

Text Prompt: A man is playing the electronic guitar, high electronic guitar.

| | Video 1 | Video 2 |
|---|---|---|
| *Text Alignment:* Which video better reflects the given text prompt? | ○ | ○ |
| *Video Quality:* Which video has better overall visual quality? | ○ | ○ |
| *Motion Fidelity:* Which video has natural and smooth motion? | ○ | ○ |

Figure A.16: **An example of human evaluation.** Participants are presented with a pair of videos and a text prompt and are instructed to evaluate text alignment, video quality, and motion fidelity.

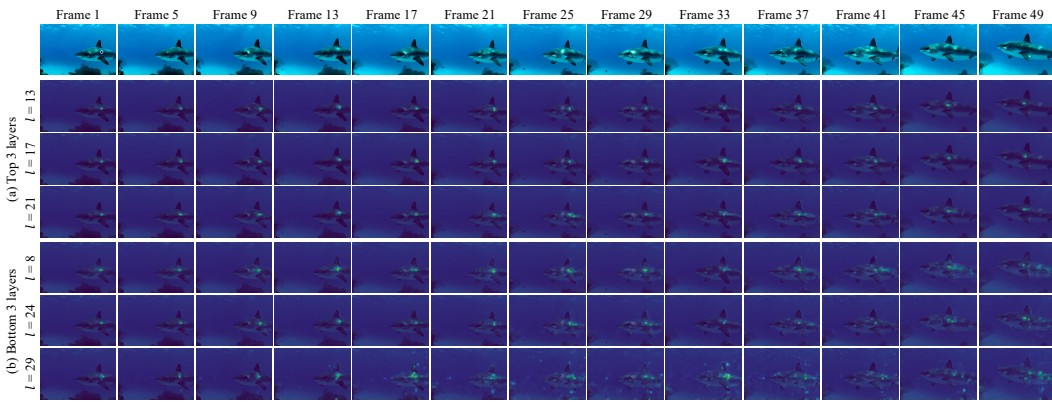

Figure A.17: **Cross-frame attention visualization across layers at timestep** $t = 1$. (a) Top-3 layers ($l = 13, 17, 21$) exhibit sharp and precise localization. (b) Bottom-3 layers ($l = 8, 24, 29$) display diffuse and scattered attention.

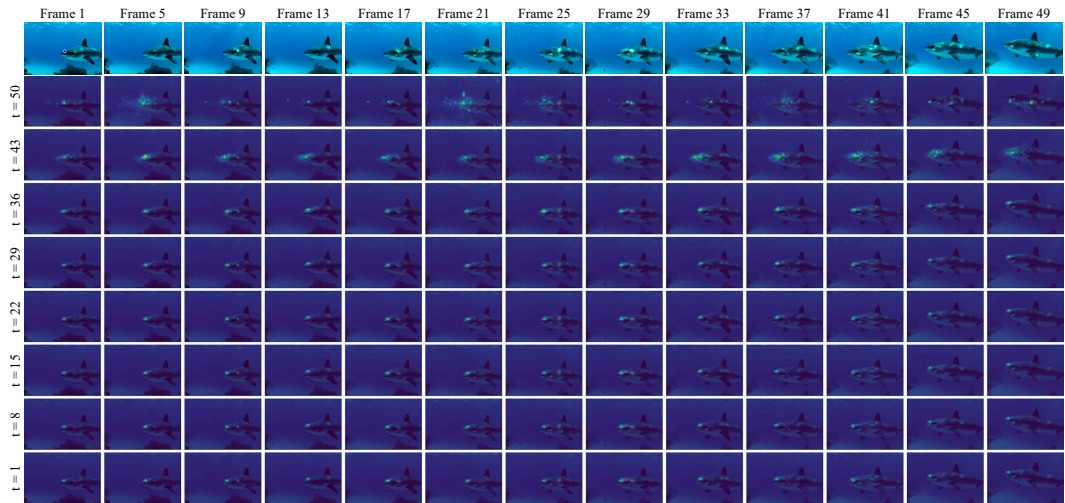

Figure A.18: **Cross-frame attention visualization across denoising timesteps at layer** $l = 17$. Attention progressively sharpens and localizes throughout denoising ($t = 50$ to 1).

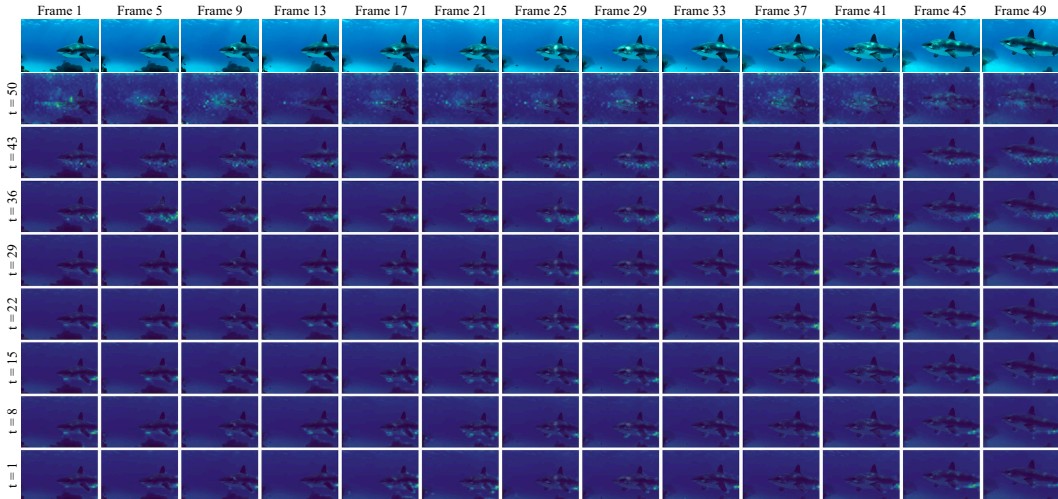

Figure A.19: **Text-to-frame attention visualization for the word "shark" across denoising timesteps at layer** $l = 17$. Attention evolves from coarse to fine throughout denoising ($t = 50$ to 1), indicating that the text prompt primarily guides the global semantic layout in early timesteps.

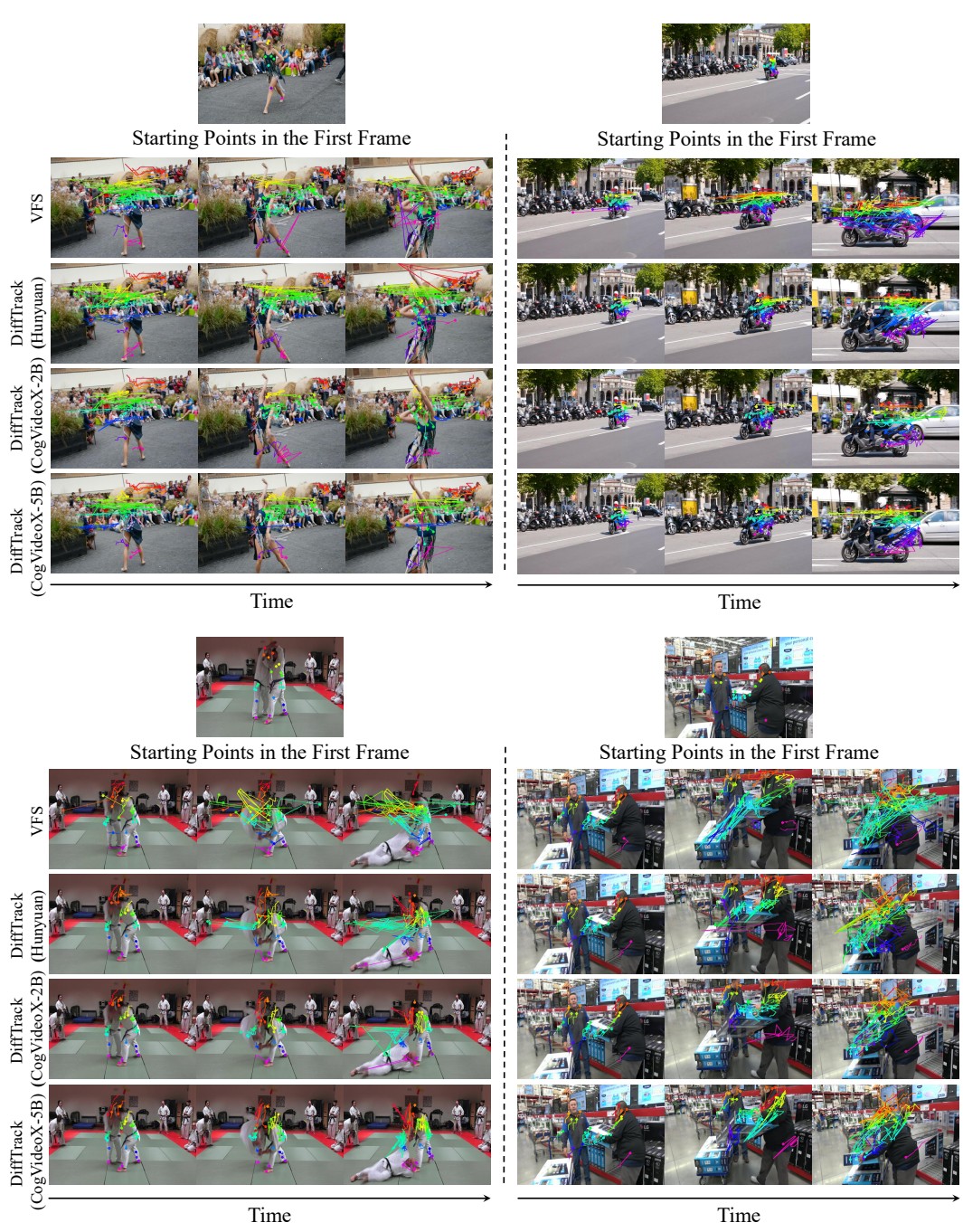

Figure A.20: **Additional qualitative comparison for zero-shot point tracking.** Our method produces smoother and more accurate trajectories compared to VFS [83], which struggle with temporal dynamics and often yield inconsistent tracks.

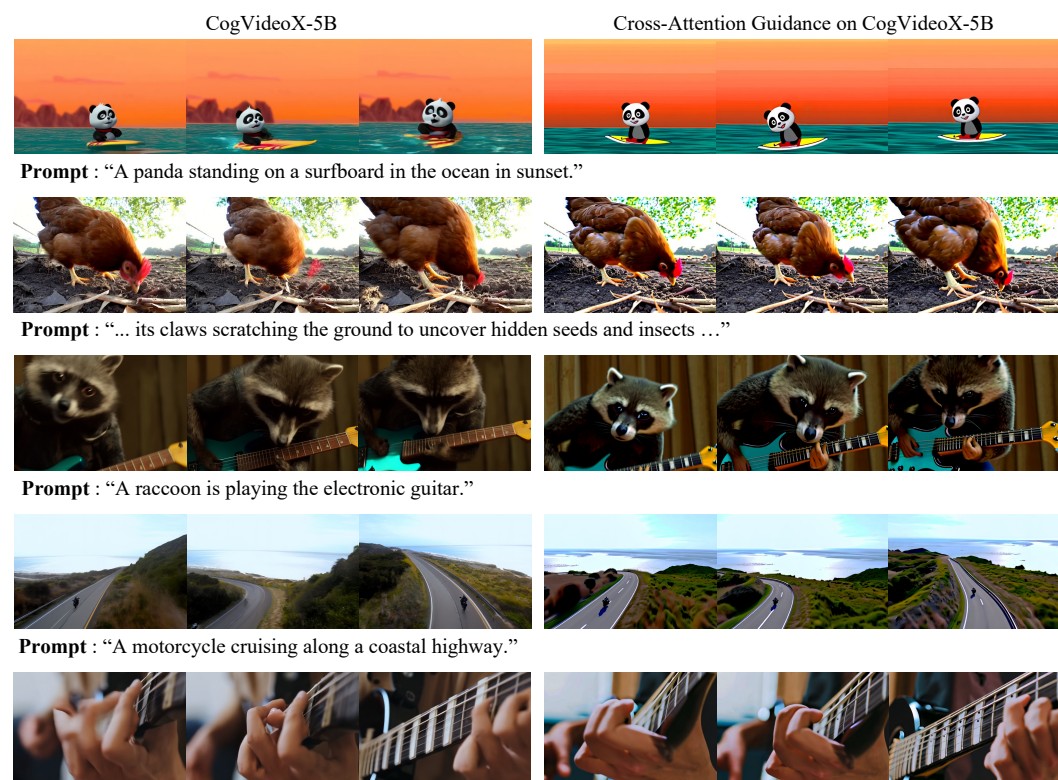

CogVideoX-5B        Cross-Attention Guidance on CogVideoX-5B

**Prompt** : "A panda standing on a surfboard in the ocean in sunset."

**Prompt** : "... its claws scratching the ground to uncover hidden seeds and insects …"

**Prompt** : "A raccoon is playing the electronic guitar."

**Prompt** : "A motorcycle cruising along a coastal highway."

**Prompt** : "A person playing guitar"

Figure A.21: **Additional qualitative results on motion-enhanced generation with CogVideoX-5B.** Our sampling method, CAG, produces videos with improved motion consistency.

## J  Broader Impact

DiffTrack advances the understanding of temporal correspondence in video diffusion transformers (DiTs), enabling applications such as zero-shot tracking and motion-enhanced video generation. These capabilities can benefit diverse downstream tasks, including point tracking [16, 43, 44], 4D point tracking [15], and motion-manipulated video generation [24].

However, DiffTrack's ability to enhance video quality may raise ethical concerns if misused to create misleading or fake content. It is essential to ensure that advancements from DiffTrack are applied responsibly, especially in video synthesis and manipulation.

## K  Limitations

DiffTrack relies on pre-trained video diffusion transformers (DiTs), meaning that advancements in video backbones could enhance its performance, resulting in more accurate tracking and coherent motion generation.

While DiffTrack effectively analyzes temporal correspondences and improves motion consistency, it does not currently support motion manipulation—direct control of video synthesis with user-defined motion trajectories. Extending DiffTrack for motion-conditioned video generation is a potential future direction.

Furthermore, DiffTrack performs zero-shot point tracking without fine-tuning on specific datasets. Incorporating fine-tuning could further improve real-world performance, which we plan to explore in future research.

