# OpenReview forum: "Emergent Temporal Correspondences from Video Diffusion Transformers"
_NeurIPS.cc/2025/Conference — NeurIPS 2025 poster_

### Official Review · Reviewer_XJK1 · 2025-06-04

**Clarity:** 4
**Significance:** 3
**Originality:** 4
**Rating:** 5
**Confidence:** 5

**Summary:**

The paper presents a study on emergent temporal correspondences in video generation using Diffusion Transformer (DiT) models.
The authors first construct a synthetic dataset by generating videos, annotating the first frame, and using CoTracker [36] to obtain pseudo ground truth trajectories.
Using this dataset, they systematically analyze how well temporal correspondences can be extracted from intermediate features and query-key similarities across different layers and denoising steps.
Based on these findings, they apply their method to zero-shot point tracking on the TAP-Vid benchmark, achieving state-of-the-art performance compared to both vision foundation models and self-supervised video point tracking methods.
In addition, they propose a novel guidance strategy that perturbs cross-frame attention to discourage degraded temporal correspondence during generation.
This approach enhances the temporal consistency of the generated videos.
Quantitative metrics and a user study demonstrate that the proposed guidance method significantly outperforms the baseline in terms of motion fidelity and overall video quality.

**Questions:**

1. Does the proposed approach generalize to object other than animals, which was the main focus of the evaluation dataset?

2. What is the inference time for the zero-shot tracker and the computational overhead from the CAG component?

3. An interesting future direction would be to explore learnable feature aggregation across timesteps and denoising levels, as proposed in [1]. Additionally, handling ambiguous matching, such as switching between different wheels of the car in the first supplementary video, could benefit from techniques like [2].

[1] Luo, Grace, et al. "Diffusion hyperfeatures: Searching through time and space for semantic correspondence." Advances in Neural Information Processing Systems 36 (2023): 47500-47510.

[2] Zhang, Junyi, et al. "Telling left from right: Identifying geometry-aware semantic correspondence." CVPR (2024)

**Ethical Concerns:**

["NO or VERY MINOR ethics concerns only"]

**Final Justification:**

The paper is clearly written and presents technical novelty supported by thorough experiments and analysis. T
he authors have addressed all of my questions and concerns during the rebuttal.
I therefore recommend acceptance.

**Limitations:**

Yes in the supplementary material, but I suggest moving it to the main paper and showing some failure cases of both point tracking and refined video generation.

**Quality:**

4

**Strengths And Weaknesses:**

**Strengths**
- The paper is very well written. The narrative is clear, smooth, and logically structured, making the problem easy to follow and well motivated.
- The preliminary section (section 2) is compact, generic, and well-structured. It effectively narrows the focus to cross-frame attention, setting up the reader for the analysis that follows.
- Section 3.4 offers valuable insights through a thorough and well-executed analysis. I believe these insights will be highly valuable for the community in understanding DiT video diffusion models.
- The CAG component for temporal refinement is creative, and has a significant impact on the quality of the generated videos.

**Weaknesses**
1. The paper lacks a dedicated related work section. Given the relevance of semantic correspondence task, it is important to discuss prior work in this area such as [1–3] to clarify what has already been achieved, as well as approaches that analyze temporal information in earlier video diffusion models like SVD [4].

2. Fig. 3(a,b) is hard to interpret. It is unclear which layer or timestep performs best. Clearer visualizations or summaries would make the findings easier to follow.

3. The object dataset was chosen to focus primarily on animals. It would be helpful to clarify the reasoning behind this choice.

4. To strengthen the evaluation, it would be beneficial to include recent diffusion-based semantic correspondence methods such as [1–3]. Additionally, adding CoTracker results to Table 1 would help position the proposed approach relative to state-of-the-art supervised methods.

5. It would be interesting to evaluate CAG on larger video models to assess whether the observed improvements persist.


[1] Luo, Grace, et al. "Diffusion hyperfeatures: Searching through time and space for semantic correspondence." Advances in Neural Information Processing Systems 36 (2023): 47500-47510.

[2] Zhang, Junyi, et al. "Telling left from right: Identifying geometry-aware semantic correspondence." CVPR (2024)

[3] Stracke, Nick, et al. "CleanDIFT: Diffusion Features without Noise." CVPR (2025).

[4] Wang, Qian, et al. "Zero-shot video semantic segmentation based on pre-trained diffusion models." CVPR (2025).

---

> ### Author Rebuttal · Authors · 2025-07-31
>
> ### **General Reply**
>
> Thank you for your constructive review and valuable suggestions. Below, we address each comment in detail. If any concerns remain, we welcome further clarification and will respond promptly. All discussions will be reflected in the final manuscript.
>
> ---
>
> ### **Weakness 1 & 4. Lack of Dedicated Related Works and More Comparisons**
>
> > The paper lacks a dedicated related work section. Given the relevance of semantic correspondence task, it is important to discuss prior work in this area such as [1–3] to clarify what has already been achieved, as well as approaches that analyze temporal information in earlier video diffusion models like SVD [4].
> >
>
> > To strengthen the evaluation, it would be beneficial to include recent diffusion-based semantic correspondence methods such as [1–3]. Additionally, adding CoTracker results to Table 1 would help position the proposed approach relative to state-of-the-art supervised methods.
> >
>
> Thank you for your feedback. First, we would like to clarify that we provide related works in Appendix H, covering prior studies on diffusion-based semantic correspondence and feature analysis in earlier video diffusion models. Following your suggestion, we will incorporate references [1, 2, 3, 4] into this section to better position our work.
>
> In the table below, we present additional comparisons with recent diffusion-based semantic correspondence methods [1–3] and CoTracker3 [5] for point accuracy on DAVIS. We observe that our method outperforms these two-image diffusion-based approaches [1-3]. We will include these results in the final manuscript.
>
> |  | $<\delta^0$ | $<\delta^1$ | $<\delta^2$ | $<\delta^3$ | $<\delta^4$ | $<\delta^x_{avg}$ |
> | --- | --- | --- | --- | --- | --- | --- |
> | CoTracker3 (Online) | 42.9 | 67.3 | 85.2 | 93.2 | 96.8 | 77.1 |
> | CoTracker3 (Offline) | 42.1 | 66.5 | 84.0 | 93.1 | 96.4 | 76.4 |
> | DIFT (SD1.5) | 3.5 | 13.0 | 39.3 | 63.1 | 72.2 | 38.2 |
> | DIFT (SD2.1) | 3.6 | 13.3 | 40.1 | 65.8 | 75.7 | 39.7 |
> | Diffusion Hyperfeatures (SD1.5) | 3.0 | 12.9 | 33.1 | 58.4 | 74.5 | 36.4 |
> | Diffusion Hyperfeatures (SD2.1) | 3.7 | 15.5 | 38.8 | 64.5 | 78.9 | 40.3 |
> | CleanDIFT (SD1.5) | 3.1 | 13.7 | 36.5 | 64.9 | 79.6 | 39.6 |
> | CleanDIFT (SD2.1) | 3.7 | 16.0 | 41.8 | 68.6 | 81.9 | 42.4 |
> | TLR (SD1.5 + DINOv2 (ViT-B/14)) | 4.3 | 17.7 | 44.6 | 69.8 | 82.0 | 43.7 |
> | CogVideoX-2B | 4.8 | 19.4 | 49.2 | 73.6 | 84.3 | 46.3 |
> | CogVideoX-5B | **5.2** | **20.5** | **50.7** | **73.9** | **84.3** | **46.9** |
>
> **References**
>
> [1] Luo, G., Dunlap, L., Park, D. H., Holynski, A., & Darrell, T. (2023). Diffusion hyperfeatures: Searching through time and space for semantic correspondence. *Advances in Neural Information Processing Systems*, *36*, 47500-47510.
>
> [2] Zhang, J., Herrmann, C., Hur, J., Chen, E., Jampani, V., Sun, D., & Yang, M. H. (2024). Telling left from right: Identifying geometry-aware semantic correspondence. In *Proceedings of the IEEE/CVF Conference on Computer Vision and Pattern Recognition* (pp. 3076-3085).
>
> [3] Stracke, N., Baumann, S. A., Bauer, K., Fundel, F., & Ommer, B. (2025). Cleandift: Diffusion features without noise. In *Proceedings of the Computer Vision and Pattern Recognition Conference* (pp. 117-127).
>
> [4] Wang, Q., Eldesokey, A., Mendiratta, M., Zhan, F., Kortylewski, A., Theobalt, C., & Wonka, P. (2024). Zero-shot video semantic segmentation based on pre-trained diffusion models. *arXiv preprint arXiv:2405.16947*.
>
> [5] Karaev, N., Makarov, I., Wang, J., Neverova, N., Vedaldi, A., & Rupprecht, C. (2024). Cotracker3: Simpler and better point tracking by pseudo-labelling real videos. *arXiv preprint arXiv:2410.11831*.
>
> ---
>
> ### **Weakness 2. Unclear Visualization**
>
> > Fig. 3(a,b) is hard to interpret. It is unclear which layer or timestep performs best. Clearer visualizations or summaries would make the findings easier to follow.
> >
>
> Thank you for your comment. We will include 2D visualizations of the per-timestep and per-layer analysis corresponding to Fig. 3(a, b) in the final manuscript to more clearly highlight the dominant layers and timesteps and reduce potential confusion. We will also add a concise summary to help readers interpret the findings more easily.
>
> ---
>
> ### **Weakness 3 & Question 1. Reason to Choose Animal Dataset**
>
> > The object dataset was chosen to focus primarily on animals. It would be helpful to clarify the reasoning behind this choice.
> >
>
> > Does the proposed approach generalize to object other than animals, which was the main focus of the evaluation dataset?
> >
>
> We chose animals for the object dataset because animal motion is typically non-rigid, dynamic, and deformable, unlike rigid objects. This results in more complex and challenging tracking scenarios, which we believe are valuable for analyzing temporal correspondence.
>
> Additionally, we include the same analysis on the DAVIS dataset (Figure A.5 in Appendix E), which contains real-world videos with dynamic human motion, further supporting our findings. We will clarify this reasoning and include it in the final version of the manuscript.
>
> ---
>
> ### **Weakness 5. Generalizability of Cross-Attention Guidance**
>
> > It would be interesting to evaluate CAG on larger video models to assess whether the observed improvements persist.
> >
>
> In Table 1 and 2 below, we evaluate the generalizability of Cross-Attention Guidance (CAG) on a larger model, CogVideoX-5B. We observe that CAG consistently outperforms the baseline across all automatic metrics and human evaluation.
>
> These results suggest that the effectiveness of CAG persists even in larger-scale models, demonstrating its scalability and robustness. We will include these quantitative results, along with additional qualitative examples, in the final version of the manuscript.
>
> Table 1. Quantitative results of CAG on CogVideoX-5B using automatic metrics on VBench.
> |  | Subject Consistency | Background Consistency | Dynamic Degree | Imaging Quality |
> | --- | --- | --- | --- | --- |
> | CogVideoX-5B | 0.9158 | 0.9590 | 0.6667 | 0.5531 |
> | CogVideoX-5B + CAG | **0.9283** | **0.9644** | **0.6863** | **0.6051** |
>
> Table 2. Quantitative results of CAG on CogVideoX-5B using human evaluation.
> |  | Video Quality | Motion Fidelity | Text Faithfulness |
> | --- | --- | --- | --- |
> | CogVideoX-5B | 39.10 | 40.91 | 30.00 |
> | CogVideoX-5B + CAG | **60.90** | **59.09** | **70.00** |
>
> ---
>
> ### **Question 2. Computational Overhead for CAG**
>
> > What is the inference time for the zero-shot tracker and the computational overhead from the CAG component?
> >
>
> The table below compares inference time and memory usage per denoising timestep across three settings: the baseline, baseline with Classifier-Free Guidance (CFG), and baseline with CFG and Cross-Attention Guidance (CAG). While inference runtime increases due to the increase of the number of function evaluations, CAG’s GPU memory footprint remains very similar to that of CFG since CAG simply applies an attention mask to a few dominant layers.
>
> This stands in contrast to prior works [1,2], which introduce additional modules and require training for motion enhancement. In comparison, our method is entirely training-free and operates within the model itself via a novel guidance mechanism.
>
> |  | Memory (GB) | Time (s) |
> | --- | --- | --- |
> | (1)  Baseline (CogVideoX-2B) | 12.75 | 1.31 |
> | (2) (1) + CFG | 12.75  | 2.60 (198% $\uparrow$) |
> | (3) (2) + CAG | 12.84 | 4.59 (177% $\uparrow$) |
>
> **References**
>
> [1] Chefer, H., Singer, U., Zohar, A., Kirstain, Y., Polyak, A., Taigman, Y., ... & Sheynin, S. (2025). Videojam: Joint appearance-motion representations for enhanced motion generation in video models. *arXiv preprint arXiv:2502.02492.*
>
> [2] Jeong, H., Huang, C. H. P., Ye, J. C., Mitra, N. J., & Ceylan, D. (2025). Track4gen: Teaching video diffusion models to track points improves video generation. In *Proceedings of the Computer Vision and Pattern Recognition Conference* (pp. 7276-7287).
>
> ---
>
> ### **Suggestion 1. Feature Aggregation across Timesteps**
>
> > An interesting future direction would be to explore learnable feature aggregation across timesteps and denoising levels, as proposed in [1]. Additionally, handling ambiguous matching, such as switching between different wheels of the car in the first supplementary video, could benefit from techniques like [2].
> >
>
> We appreciate your valuable suggestion. We agree that a promising direction for DiffTrack is to explore learnable feature aggregation across different timesteps and layers [1] to further enhance temporal correspondence.
>
> We also find your point about handling ambiguous or noisy matches insightful, for example, the issue of switching between similar keypoints like car wheels. As you suggested, adopting more advanced matching techniques such as [2] could be an effective solution.
>
> Thank you again for your thoughtful feedback. We plan to actively investigate these directions in our future work.
>
> **References**
>
> [1] Luo, G., Dunlap, L., Park, D. H., Holynski, A., & Darrell, T. (2023). Diffusion hyperfeatures: Searching through time and space for semantic correspondence. *Advances in Neural Information Processing Systems*, *36*, 47500-47510.
>
> [2] Zhang, J., Herrmann, C., Hur, J., Chen, E., Jampani, V., Sun, D., & Yang, M. H. (2024). Telling left from right: Identifying geometry-aware semantic correspondence. In *Proceedings of the IEEE/CVF Conference on Computer Vision and Pattern Recognition* (pp. 3076-3085).
>
> ---
>
> ### **Suggestion 2. Limitations**
>
> > Yes in the supplementary material, but I suggest moving it to the main paper and showing some failure cases of both point tracking and refined video generation.
> >
>
> Thank you for the comment. We will move the limitations section into the main paper in the final version and provide additional failure cases of point tracking and video generation.

---

> > ### Comment · Reviewer_XJK1 · 2025-08-05
> >
> > Thank you for addressing my comments and questions. Great work.

---

> ### Author Response · Authors · 2025-08-05
>
> Dear Reviewer XJK1,
>
> Thank you for your positive feedback and for recognizing our efforts in addressing your comments.
>
> If you believe our discussions have improved the paper, we would appreciate it if you could reflect that in your final score.
>
> Best regards,
>
> The Authors of Paper 16567

---

### Official Review · Reviewer_YjNJ · 2025-07-02

**Clarity:** 4
**Significance:** 3
**Originality:** 2
**Rating:** 5
**Confidence:** 4

**Summary:**

This paper introduces DiffTrack, a framework designed to analyze the underlying temporal correspondences between frames (thereby point tracking) during the generation process of video diffusion transformers (DiTs). The paper shows results probing the internal mechanisms of DiTs, investigating which representations (e.g. query-key similarities vs. intermediate features), layers, and timesteps are most critical for tracking. Their key findings are that query-key similarities in the attention mechanism are the primary source of correspondence information, and the tracking quality improves as the denoising process progresses. With these insights, the paper presents two practical applications. First, it shows that video DiTs can be used as zero-shot point trackers, achieving results comparable to existing tracking methods. Second, it introduces a training-free guidance method to enhance motion quality in the generated video, by perturbing the cross-frame attention maps in some key layers to steer the model towards generating videos with fewer artifacts.

**Questions:**

- How well will the findings (e.g. layer selection) generalize across different text prompts?
- Will different content types lead to different layers that provide strong correspondence information?
- Using the proposed method on CogVideoX-2B-I2V, the performance is actually a lot worse than SVD, which is also an I2V model. Can you comment on this?

**Ethical Concerns:**

["NO or VERY MINOR ethics concerns only"]

**Final Justification:**

The author response has addressed my concerns.

**Limitations:**

Yes

**Quality:**

3

**Strengths And Weaknesses:**

Strengths
- The paper successfully translates analytical findings about the inner mechanisms of video DiTs into concrete applications like tracking and video quality enhancement.
- The paper is exceptionally well-written and easy to follow. The concepts are explained clearly.
- The analysis is systematic and provides clear and valuable insights into the behavior of video DiTs.

Weaknesses
- The central discovery that attention maps in diffusion models capture spatial correspondences is not entirely novel. Lots of prior works found this phenomenon in image diffusion models. This paper's contribution is to verify this unsurprising finding in the context of video DiTs.
- There's a quite significant gap between the tracking performance of CogVideoX and CogVideoX-I2V, which is revealed in the appendix. It would be more appropriate to discuss this in the main paper.
- Table 1's message is that CogVideoX combined with the proposed DiffTrack "outperform all vision foundation models and self-supervised video models". However, when CogVideoX-I2V is combined with the proposed DiffTrack, its performance is worse than all other baseline methods, including SVD which is an image-to-video model. These results seem to indicate that the proposed DiffTrack achieves mixed results.

---

> ### Author Rebuttal · Authors · 2025-07-31
>
> ### **General Reply**
>
> Thank you for your constructive review and valuable suggestions. Below, we address each comment in detail. If any concerns remain, we welcome further clarification and will respond promptly. All discussions will be reflected in the final manuscript.
>
> ---
>
> ### **Weakness 1. Novelty of DiffTrack**
>
> > The central discovery that attention maps in diffusion models capture spatial correspondences is not entirely novel. Lots of prior works found this phenomenon in image diffusion models. This paper's contribution is to verify this unsurprising finding in the context of video DiTs.
> >
>
> Thank you for the comment. We would like to highlight that reviewers Enkm, vZ2J, and YjNJ acknowledged the novelty in both our analysis (Enkm, vZ2J) and applications (vZ2J, XJK1).
>
> DiffTrack is the first work to quantify temporal correspondence in full 3D attention within video DiTs. As you mentioned, previous works [1,2,3] have explored intermediate features from pre-trained image diffusion models for correspondence. However, their analyses are primarily limited to two-frame correspondence, as image diffusion models are not designed to capture temporal relationships across video sequences.
>
> Recent works [4,5] have attempted to analyze internal representations of video diffusion models for controlled video generation. However, they do not explicitly study or quantify temporal correspondence, and are typically based on U-Net architectures, which are known to struggle with large motion. Another recent study [6] investigates attention control in video DiTs for subject consistency in long video generation, but it focuses solely on text-to-video attention, overlooking temporal alignment between frames.
>
> In contrast, our work systematically quantifies how video DiTs build temporal correspondence across frames during video generation, using novel evaluation metrics and systems. We show how these temporal signals can be extracted (zero-shot point tracking) and leveraged for motion-enhanced video generation (Cross-Attention Guidance).
>
> We believe our work offers crucial insights into the inner workings of video DiTs, and establishes a strong foundation for future research and applications that aim to leverage their temporal understanding.
>
> **References**
>
> [1] Hedlin, E., Sharma, G., Mahajan, S., Isack, H., Kar, A., Tagliasacchi, A., & Yi, K. M. (2023). Unsupervised semantic correspondence using stable diffusion. *Advances in Neural Information Processing Systems*, *36*, 8266-8279.
>
> [2] Meng, B., Xu, Q., Wang, Z., Cao, X., & Huang, Q. (2024). Not all diffusion model activations have been evaluated as discriminative features. *Advances in Neural Information Processing Systems*, *37*, 55141-55177.
>
> [3] Tang, L., Jia, M., Wang, Q., Phoo, C. P., & Hariharan, B. (2023). Emergent correspondence from image diffusion. *Advances in Neural Information Processing Systems*, *36*, 1363-1389.
>
> [4] Jeong, H., Huang, C. H. P., Ye, J. C., Mitra, N. J., & Ceylan, D. (2025). Track4gen: Teaching video diffusion models to track points improves video generation. In Proceedings of the Computer Vision and Pattern Recognition Conference (pp. 7276-7287).
>
> [5] Xiao, Z., Zhou, Y., Yang, S., & Pan, X. (2024). Video diffusion models are training-free motion interpreter and controller. Advances in Neural Information Processing Systems, 37, 76115-76138.
>
> [6] Cai, M., Cun, X., Li, X., Liu, W., Zhang, Z., Zhang, Y., ... & Yue, X. (2025). Ditctrl: Exploring attention control in multi-modal diffusion transformer for tuning-free multi-prompt longer video generation. In *Proceedings of the Computer Vision and Pattern Recognition Conference* (pp. 7763-7772).
>
> ---
> ### **Weakness 2, 3 & Question 3. Performance of Image-to-Video Models**
>
> > There's a quite significant gap between the tracking performance of CogVideoX and CogVideoX-I2V, which is revealed in the appendix. It would be more appropriate to discuss this in the main paper.
> >
>
> > Table 1's message is that CogVideoX combined with the proposed DiffTrack "outperform all vision foundation models and self-supervised video models". However, when CogVideoX-I2V is combined with the proposed DiffTrack, its performance is worse than all other baseline methods, including SVD which is an image-to-video model. These results seem to indicate that the proposed DiffTrack achieves mixed results.
> >
>
> > Using the proposed method on CogVideoX-2B-I2V, the performance is actually a lot worse than SVD, which is also an I2V model. Can you comment on this?
> >
>
> Thank you for your valuable comments.
>
> Regarding CogVideoX-2B-I2V, we used unofficial pre-trained weights from a third-party GitHub repository, with unknown training strategy and dataset. Notably, this model concatenates the first condition image to all frame latents, unlike the official I2V setting, which concatenates it only to the first latent while zero-padding the rest. This might lead to a significant drop in performance, as the model over-relies on the first image and ultimately produces largely static videos. We also suspect this low performance stems from the model’s undocumented training procedure or data, which warrants further investigation.
>
> For rigorous analysis, in the table below, we further evaluate the officially released CogVideoX-I2V-5B on DAVIS and observe that it outperforms SVD in point tracking accuracy.
>
> We also observe CogVideoX-5B-I2V underperform compared to their T2V counterpart CogVideoX-5B. We believe this is due to the image-to-video finetuning objective, which primarily focuses on preserving the appearance of the first frame, producing more static videos and weakening the temporal correspondence required for generating dynamic motion.
>
> This phenomenon is also discussed in [1], which states:
>
> *“Adapting a T2V model for I2V suppresses motion dynamics of the generated outputs, resulting in more static videos compared to their T2V counterparts... It stems from the premature exposure to high-frequency details in the input image, which biases the sampling process toward a shortcut trajectory that overfits to the static appearance of the reference image.”*
>
> We will incorporate this discussion more explicitly into our main paper.
>
> |  | $<\delta^0$ | $<\delta^1$ | $<\delta^2$ | $<\delta^3$ | $<\delta^4$ | $<\delta^x_{avg}$ |
> | --- | --- | --- | --- | --- | --- | --- |
> | SVD | 3.6 | 14.6 | 34.1 | 55.7 | 71.4 | 35.9 |
> | CogVideoX-5B-I2V | 3.9 | 16.0 | 38.2 | 57.0 | 69.0 | 36.8 |
> | CogVideoX-5B | 5.2 | 20.5 | 50.7 | 73.9 | 84.3 | 46.9 |
>
> **References**
>
> [1] Choi, J. S., Lee, K., Yu, S., Choi, Y., Shin, J., & Lee, K. (2025). Enhancing Motion Dynamics of Image-to-Video Models via Adaptive Low-Pass Guidance. *arXiv preprint arXiv:2506.08456*.
>
> ---
>
> ### **Question 1 & 2. Generalizability across Different Text Prompts and Contents**
>
> > How well will the findings (e.g. layer selection) generalize across different text prompts?
> >
>
> > Will different content types lead to different layers that provide strong correspondence information?
> >
>
> Thank you for your valuable question. We would like to clarify that, to find generalizable timesteps and layers for temporal correspondence across different prompts and contents, we intentionally curated two distinct datasets: an object dataset for dynamic object-centric videos and a scene dataset for static scenes with different camera motions (lines 131–134). Each dataset includes 50 diverse prompts, covering a wide range of content types such as animals, objects, architectures, and landscapes.
>
> We observed that the same timesteps and layers are dominant across both datasets, indicating consistent temporal correspondence behavior regardless of text prompts or content types.
>
> This observation is further supported by Appendix Figure A.5, where we conducted the same analysis on the DAVIS dataset, which contains real videos with dynamic human motion. The same dominant layers and timesteps emerged, strongly aligning with our findings from synthetic videos.

---

> > ### Comment · Reviewer_YjNJ · 2025-08-06
> >
> > Thank you for the response. I will update the rating to accept.

---

> > > ### Author Response · Authors · 2025-08-06
> > >
> > > Dear Reviewer YjNJ,
> > >
> > > Thank you for your constructive discussion and your update.
> > >
> > > Best regards,
> > >
> > > The Authors of Paper 16567

---

### Official Review · Reviewer_vZ2J · 2025-07-03

**Clarity:** 4
**Significance:** 2
**Originality:** 2
**Rating:** 4
**Confidence:** 3

**Summary:**

This work designs a novel way for identifying which intermediate variables are most relevant to temporal correspondences and leveraging them to solve downstream tasks: point tracking and motion-enhanced video generation. Empirical results verify the effectiveness of proposed method.

**Questions:**

1. I am wondering how the author leveraged the existing vision foundation and supervised video models to handle the zero-shot point tracking task. Does the author also construct the track based on the query-key attention scores of specific layers? Please clarify it to prove the comparison is fair.
2. The author only compares to the vanilla model, but there should be other relevant works for the motion-enhanced video generation task. Please provide their results to further verify the effectiveness of the method.

**Ethical Concerns:**

["NO or VERY MINOR ethics concerns only"]

**Final Justification:**

The authors' response have clarified my concerns. I hope the authors can refine their final paper accordingly.

**Limitations:**

Yes.

**Paper Formatting Concerns:**

No formatting concerns.

**Quality:**

3

**Strengths And Weaknesses:**

Strength:
1. The paper is generally easy to follow.
2. Detailed experiments on three video generation models (CogVideoX-2B ，HunyuanVideo and CogVideoX-2B-I2V) are presented.

Weakness:
1. The fairness and integrity of the comparison of experiments need to further verify. For point tracking task, the implementation of baseline models is not stated. For motion-enhanced video generation task, the results were only compared to the vanilla model.
2. Given that the best timestep is not always the last one, the generality of the proposed point tracking method needs further validation. For example, DiffTrack for zero-shot point tracking  using the last timestep. While according to the appendix, for CogVideoX-2B-I2V, the best timestep is not the last. This can undermine point tracking performance. The author need to either prove the best timestep is the last for most video generation models or prove using the  non-last timesteps can also achieve good point tracking performance.

---

> ### Author Rebuttal · Authors · 2025-07-31
>
> ### **General Reply**
>
> Thank you for your constructive review and valuable suggestions. Below, we address each comment in detail. If any concerns remain, we welcome further clarification and will respond promptly. All discussions will be reflected in the final manuscript.
>
> ---
>
> ### **Weakness 1 & Question 1. Implementation of Comparison Models**
>
> > The fairness and integrity of the comparison of experiments need to further verify. For point tracking task, the implementation of baseline models is not stated.
> >
>
> > I am wondering how the author leveraged the existing vision foundation and supervised video models to handle the zero-shot point tracking task. Does the author also construct the track based on the query-key attention scores of specific layers? Please clarify it to prove the comparison is fair.
> >
>
> We kindly note that implementation details for all comparison models are provided in Appendix C.3.
>
> For vision foundation models such as DINO, DINOv2, DINOv2-Reg, and DIFT (SD 1.5 and SD 2.1), we strictly followed the protocols from [1], which benchmark zero-shot point tracking performance across vision foundation models. Since Stable Diffusion (SD 1.5, SD 2.1) does not provide optimal feature descriptors for correspondence, we followed DIFT [2], consistent with [1], and used features from the third upsampling block.
>
> For self-supervised video models (SMTC, CRW, SPA-the-Temp, VFS), we also used their official implementations as feature extractors. For Stable Video Diffusion (SVD), since the official implementation does not specify an optimal feature for matching, we followed [3] and extracted features from the third upsampling block. The aforementioned models do not use query-key features for matching. For ZeroCo, we used query-key matching from the cross-attention map, which is their default matching setting.
>
> Following [1], since different models produce features at varying spatial resolutions, we first downsampled the input frames to 256 × 256, and then resized them such that all output feature maps have a uniform resolution of 30 × 45 before computing correlations.
>
> We will expand this explanation in the final manuscript and publicly release the evaluation code including all comparison baselines.
>
>
> **References**
>
> [1] Aydemir, G., Xie, W., & Güney, F. (2024). Can Visual Foundation Models Achieve Long-term Point Tracking?. arXiv preprint arXiv:2408.13575.
>
> [2] Tang, L., Jia, M., Wang, Q., Phoo, C. P., & Hariharan, B. (2023). Emergent correspondence from image diffusion. *Advances in Neural Information Processing Systems*, *36*, 1363-1389.
>
> [3] Jeong, H., Huang, C. H. P., Ye, J. C., Mitra, N. J., & Ceylan, D. (2025). Track4gen: Teaching video diffusion models to track points improves video generation. In *Proceedings of the Computer Vision and Pattern Recognition Conference* (pp. 7276-7287).
>
> ---
>
> ### **Weakness 2. Best Timestep for Point Tracking**
>
> > Given that the best timestep is not always the last one, the generality of the proposed point tracking method needs further validation. For example, DiffTrack for zero-shot point tracking using the last timestep. While according to the appendix, for CogVideoX-2B-I2V, the best timestep is not the last. This can undermine point tracking performance. The author need to either prove the best timestep is the last for most video generation models or prove using the non-last timesteps can also achieve good point tracking performance.
> >
>
> Thank you for your insightful comment. Regarding CogVideoX-2B-I2V, we used unofficial pre-trained weights from a third-party GitHub repository, with unknown training strategy and dataset. Notably, this model concatenates the first condition image to all frame latents, unlike the official I2V setting, which concatenates it only to the first latent while zero-padding the rest. This often causes the model to over-rely on the first image and produce largely static videos, which may explain the highest matching accuracy at timestep $t = 11$. We also think such misalignment stems from the model’s unknown training strategy or data and warrants further investigation.
>
> For rigorous analysis, we further evaluated the officially released CogVideoX-5B-I2V model. In this case, we consistently observed that the last timestep yields the highest matching accuracy. In the table below, we show its point tracking performance of CogVideoX-5B-I2V ($l=16$, $t=1$) on DAVIS, revealing a significant performance gap compared to the unofficial 2B model. We appreciate your careful observation and will clarify this point in the revised version.
>
> |  | $<\delta^0$ | $<\delta^1$ | $<\delta^2$ | $<\delta^3$ | $<\delta^4$ | $<\delta^x_{avg}$ |
> | --- | --- | --- | --- | --- | --- | --- |
> | CogVideoX-2B-I2V (Unofficial) | 0.8 | 6.0 | 13.3 | 26.5 | 44.7 | 18.3 |
> | CogVideoX-5B-I2V (Official) | **3.9** | **16.0** | **38.2** | **57.0** | **69.0** | **36.8** |
>
> ---
>
> ### **Weakness 1 & Question 2. Motion-Enhanced Video Generation**
>
> > For motion-enhanced video generation task, the results were only compared to the vanilla model.
> >
>
> > The author only compares to the vanilla model, but there should be other relevant works for the motion-enhanced video generation task. Please provide their results to further verify the effectiveness of the method.
> >
>
> Thank you for your valuable suggestion.
>
> While prior works on motion-enhanced video generation [1,2] typically rely on additional supervision or auxiliary modules, our method specifically focuses on enhancing motion without any supervision or auxiliary modules, enabling a more general and scalable solution.
>
> To strengthen our evaluation, we compare our Cross-Attention Guidance (CAG) with Spatiotemporal Skip Guidance (STG) [3], which also aims to enhance motion without training. STG degrades the original model through selective skipping of spatiotemporal layers (including self-frame and text-frame attention), and then uses the degraded model as guidance.
>
> As shown in the table below, CAG outperforms STG across motion consistency, motion dynamics and video quality. While STG modifies self-frame and text-frame attention, often altering the overall layout, structure, and content of the generated videos, CAG only zeros out cross-frame attention in dominant layers. This allows CAG to enhance motion while preserving the original layout and content of the generated video.
>
> Moreover, STG does not analyze how temporal correspondence emerges within video DiTs, and therefore suffers from the following limitation, as stated in their paper:
>
> *“STG’s performance depends on scale and layer selection, with the optimal configuration varying across models, requiring users to set these through heuristic tuning.”*
>
> In contrast, CAG benefits from our DiffTrack-based analysis, which identifies optimal layers for motion-enhanced guidance, resulting in consistent performance without any manual tuning by users.
>
> We will include this comparison and discussion in the final manuscript to better clarify the advantages and positioning of our approach.
>
> |  | Subject Consistency | Background Consistency | Dynamic Degree | Imaging Quality |
> | --- | --- | --- | --- | --- |
> | (1) CogVideoX-2B | 0.9276 | 0.9490 | 0.7917 | 0.5657 |
> | (2) (1) + STG | 0.9263 | 0.9507 | 0.7777 | 0.6031 |
> | (3) (1) + CAG | **0.9313** | **0.9564** | **0.8235** | **0.6054** |
>
> **References**
>
> [1] Chefer, H., Singer, U., Zohar, A., Kirstain, Y., Polyak, A., Taigman, Y., ... & Sheynin, S. (2025). Videojam: Joint appearance-motion representations for enhanced motion generation in video models. *arXiv preprint arXiv:2502.02492.*
>
> [2] Jeong, H., Huang, C. H. P., Ye, J. C., Mitra, N. J., & Ceylan, D. (2025). Track4gen: Teaching video diffusion models to track points improves video generation. In *Proceedings of the Computer Vision and Pattern Recognition Conference* (pp. 7276-7287).
>
> [3] Hyung, J., Kim, K., Hong, S., Kim, M. J., & Choo, J. (2025). Spatiotemporal skip guidance for enhanced video diffusion sampling. In *Proceedings of the Computer Vision and Pattern Recognition Conference* (pp. 11006-11015).

---

> ### Comment · Reviewer_vZ2J · 2025-08-06
> **Re: rebuttal**
>
> Thanks for the authors' detailed response, which has clarified most of my concerns. I will raise my score.

---

> > ### Author Response · Authors · 2025-08-06
> >
> > Dear Reviewer vZ2J,
> >
> > Thank you for your update and for acknowledging our efforts in addressing your feedback.
> >
> > Best regards,
> >
> > The Authors of Paper 16567

---

### Official Review · Reviewer_EnKm · 2025-07-07

**Clarity:** 4
**Significance:** 3
**Originality:** 3
**Rating:** 5
**Confidence:** 4

**Summary:**

The paper proposes to systematically analyze transformer-based video diffusion models (DiTs) to uncover the mechanism responsible for preserving the temporal correspondence. The authors propose a way to measure temporal correspondence through key-point tracking, and create a dataset of prompt-videos to do so. This leads to the finding that key-query 3D attention captures the temporal consistency in the generation. The authors propose to use this property to develop a SoTA video keypoint tracking approach and to improve temporal consistency of the generated videos with guidance.

**Questions:**

Questions/Suggestions:
- It feels like the zero-shot point tracking is a separate contribution not aligned with the main message of the paper. Perhaps the authors may want to consider removing or rephrasing it, to make the paper more to the point.
- Long sequence handling (line 249): each chunk contains the first global frame. How does that affect 3D VAE (or is it done after) and how does this affect the 3D attention?

**Ethical Concerns:**

["NO or VERY MINOR ethics concerns only"]

**Final Justification:**

I thank the authors for answering my questions. I keep my score and vote for accepting the paper.

**Quality:**

3

**Strengths And Weaknesses:**

Strengths:
- The paper is really well written and easy to follow
- The idea of analyzing both affinity and matching accuracy is grerat. One is indeed not sufficient.
- The results in both video generation and point tracking are impressive.
- I really like the pipeline for video generation with negative guidance.

Weaknesses:
- My biggest concern is the prompt-video dataset. To establish the dataset, the authors generate videos with a text-to-video diffusion model. Then, to get the annotations, apply SAM and Co-Tracker to the generated videos. I have two questions here:
    - The SAM and Co-tracker models were trained primarily on real videos. How is the accuracy of those models on the generated videos? Does this introduce an error? I find this gap potentially concerning. Given that some of the generated objects’ appearance may drift over time due to imperfect generation, what would it even mean to match the new halluzitated keypoints to the original frame?
    - By generating videos from a specific generative model (e.g., CogVideo), doesn’t it make the dataset somewhat biased for evaluating other video generative models?

---

> ### Author Rebuttal · Authors · 2025-07-31
>
> ### **General Reply**
>
> Thank you for your constructive review and valuable suggestions. Below, we address each comment in detail. If any concerns remain, we welcome further clarification and will respond promptly. All discussions will be reflected in the final manuscript.
>
> ---
>
> ### **Weakness 1. Performance of SAM and Co-Tracker on Synthetic Datasets**
>
> > The SAM and Co-tracker models were trained primarily on real videos. How is the accuracy of those models on the generated videos? Does this introduce an error? I find this gap potentially concerning. Given that some of the generated objects’ appearance may drift over time due to imperfect generation, what would it even mean to match the new halluzitated keypoints to the original frame?
> >
>
> Thank you for your question. As noted in Appendix A (line 342), we first collect 300 prompts for each object and scene dataset. Human annotators then carefully select the top 50 video–prompt pairs per dataset based on motion consistency and overall video fidelity. This manual selection process excludes generated samples that exhibit hallucination or significant appearance drift, which may cause errors in SAM or CoTracker.
>
> Additionally, as shown in Appendix E (Figure A.5), we perform the same analysis on real videos from the DAVIS dataset, which are free from synthetic artifacts that could affect the performance of SAM or CoTracker. The results show consistent trends with those from the curated synthetic datasets, further supporting the robustness of our analysis. We will include this discussion in our final manuscript.
>
> ---
>
> ### **Weakness 2. Potential Bias from Using a Dataset Generated by a Specific Model**
>
> > By generating videos from a specific generative model (e.g., CogVideo), doesn’t it make the dataset somewhat biased for evaluating other video generative models?
> >
>
> Thank you for your question. We would like to clarify that we use a different synthetic dataset for each video generative model to avoid bias in the evaluation (lines 126–128 in the main paper). Specifically, we curate video–prompt pairs separately for each generative backbone and use them only for analyzing that corresponding model. We will revise the paper to make this point clearer and prevent potential confusion.
>
> ---
>
> ### **Suggestion 1. Clarifying the Role of Zero-Shot Point Tracking**
>
> > It feels like the zero-shot point tracking is a separate contribution not aligned with the main message of the paper. Perhaps the authors may want to consider removing or rephrasing it, to make the paper more to the point.
> >
>
> Thank you for your suggestion. The main goal of our paper is to understand how temporal correspondences emerge in video DiTs, and how these correspondences can be extracted (via zero-shot point tracking) and leveraged (for motion-enhanced video generation). In this context, the zero-shot point tracking experiments provide concrete evidence that the extracted correspondences from video DiTs are meaningful and superior to those from existing vision foundation models and self-supervised video models.
>
> In the revised version, we will clarify the connection between our core message and the zero-shot tracking experiment to avoid confusion about its role in the overall contribution.
>
> ---
> ### **Question 1. Long Sequence Handling**
>
> > Each chunk contains the first global frame. How does that affect 3D VAE (or is it done after) and how does this affect the 3D attention?
> >
>
> Thank you for your question. To clarify, the inclusion of the global first frame in each chunk is done after the 3D VAE encoding.
>
> As described in Appendix C.1 and Figure A.4, we first pass all video frames individually through the pre-trained 3D VAE with a temporal compression ratio of $q=1$ (Figure A.4(a)). We then construct each chunk by inserting the global first-frame latent (Figure A.4(b)) alongside interleaved subsequent frame latents (Figure A.4(c)).
>
> In the 3D attention, all frame latents in each chunk interact with each other through cross-frame attention. We then extract queries and keys from the optimal layer and timestep identified by DiffTrack to compute matching costs between the first frame’s query and all other frames’ keys within each chunk (Eq. (5)).
>
> As demonstrated in Table A.2 of Appendix F, our design choice avoids VAE compression loss, ensures direct matching between the global first frame and subsequent frames, and leverages cross-frame interaction in 3D attention, resulting in significantly improved point accuracy for long-term tracking.
>
> We will make this workflow clearer in the revised version.

---

> > ### Comment · Reviewer_EnKm · 2025-08-08
> > **Final thoughts**
> >
> > I thank the authors for providing a detailed response. This has clarified my  questions and concerns. I keep my score and vote for accepting the paper.

---

> ### Author Response · Authors · 2025-08-05
>
> Dear Reviewer EnKm,
>
> Thank you for finalizing your review. If you feel that our responses and clarifications have improved the paper, we would sincerely appreciate it if you could reflect that in your final score.
>
> If there are any remaining concerns or unresolved points, we would be glad to further address them.
>
> Best regards,
>
> The Authors of Paper 16567

---

### Author Response · Authors · 2025-08-05

Dear Reviewers Enkm, vZ2J, and YjNJ,

Thank you for the time and effort you have dedicated to reviewing our paper. We have carefully considered your feedback and addressed all questions and concerns. As we approach the final stages of this process, we welcome any additional comments or suggestions you may have.

Best regards,

The Authors of Paper 16567

---

### Note · Authors · 2025-08-12

We sincerely thank all reviewers (EnKm, vZ2J, YjNJ, XJK1) for their positive ratings and constructive feedback, as well as the AC for their time and effort. We are grateful that our rebuttal addressed your concerns and that you found the contributions of our work valuable.

Our paper presents a systematic analysis of temporal correspondence in video diffusion transformers (DiTs), quantifies how such signals emerge in full 3D attention, and demonstrates how they can be effectively extracted (zero-shot point tracking) and leveraged for motion-enhanced video generation (Cross-Attention Guidance). We appreciate that the reviewers recognized both the novelty of our analysis and the practical impact of our applications.

In the final manuscript, we commit to the following improvements based on our rebuttal discussion:

- (EnKm) Clarify robustness to potential error drift from SAM/CoTracker through detailed synthetic-data curation, and describe how we avoid bias from specific generative models.

- (EnKm) Clarify the role of zero-shot point tracking and its alignment with our main contributions.

- (vZ2J) Expand baseline implementation details to ensure fair comparisons, and add STG results to strengthen the evaluation of motion-enhanced generation.

- (vZ2J, YjNJ) Address the issue with the unofficial CogVideoX-2B-I2V model by additionally evaluating the official CogVideoX-5B-I2V, clarify that our best-timestep analysis is generally applicable, and further discuss behavioral differences between T2V and I2V models.

- (YjNJ) Clarify our diverse dataset curation and real-world evaluations to demonstrate the generalizability of our analysis.

- (YjNJ, XJK1) Add a dedicated Related Work section to clearly position our contribution and novelty.

- (XJK1) Include comparisons with recent diffusion-based correspondence methods and CoTracker, demonstrate CAG scalability on larger models, improve visualizations, and clarify the choice of the animal dataset.

We believe these revisions address all review points and further strengthen our contributions. This work provides new insights into the temporal reasoning capabilities of video DiTs and offers practical tools for analysis, point tracking, and video generation.

Once again, we sincerely thank the reviewers and AC for their constructive engagement, which has greatly improved the clarity and quality of our work.

---

### Decision · Program_Chairs · 2025-09-17

**Decision:**

Accept (poster)

**Comment:**

After the rebuttal, all reviewers confirmed that the issues raised have been effectively addressed and expressed consistent support for this submission. The manuscript is well-written, and the proposed method is both interesting and novel. I recommend its acceptance. I encourage the authors to enhance the camera-ready version based on the feedback provided.